# Double-Bounded Nonlinear Optimal Transport for Size Constrained Min Cut Clustering

## Abstract

Min cut is an important graph partitioning method. However, current solutions to the min cut problem suffer from slow speeds, difficulty in solving, and often converge to simple solutions. To address these issues, we relax the min cut problem into a double-bounded constraint and, for the first time, treat the min cut problem as a double-bounded nonlinear optimal transport problem. Additionally, we develop a method for solving d-bounded nonlinear optimal transport based on the Frank-Wolfe method (abbreviated as DNF). Notably, DNF not only solves the size constrained min cut problem but is also applicable to all double-bounded nonlinear optimal transport problems. We prove that for convex problems satisfying Lipschitz smoothness, the DNF method can achieve a convergence rate of $\mathcal{O}(\frac{1}{t})$. We apply the DNF method to the min cut problem and find that it achieves state-of-the-art performance in terms of both the loss function and clustering accuracy at the fastest speed, with a convergence rate of $\mathcal{O}(\frac{1}{\sqrt{t}})$. Moreover, the DNF method for the size constrained min cut problem requires no parameters and exhibits better stability. Our Code in Appendix B.3.

## 1 Introduction

Graph clustering is a fundamental issue in machine learning, widely applied in diverse fields such as computer vision (Yan et al., 2024), gene analysis (Liu et al., 2024), social network analysis (Singh et al., 2024) and many others. Among the numerous graph clustering approaches, Min Cut clustering (MC) stands out as a classical method (Henzinger et al., 2024). Despite its effectiveness, the MC problem has trivial solution that all the objects are clustered into one cluster, where the cut of $G$ reaches it minimal value zero. Thus, it is known that the clustering result of MC tends to produce unbalanced clusters, often resulting in small, fragmented groups due to its tendency to prioritize cuts with minimal edge weights (Nie et al., 2010).

To address this limitation, various refinements to MC have been proposed (Hagen & Kahng, 1992; Zhong & Pun, 2021; Tsitsulin et al., 2023). Recently, (Nie et al., 2024) propose the parameter-insensitive min cut clustering with flexible size constraints. In fact, the most direct approach to balance clustering results in MC is to add size constraints for each cluster. That is, in MC problem, the lower bound $b_l$ and upper bound $b_u$ are added in column sums of discrete indicator matrix $Y$, which guarantees each clusters contains reasonable number of objects. Nevertheless, the optimization for the size constrained problem is not easy since the coupling of constraints. For each row of $Y$, it is required that only one element is one and others are zeros. The sum of column needs in the range of $[b_l, b_u]$. In this paper, we relax the discrete indicator matrix into probabilistic constraints and resolve this problem from the perspective of nonlinear optimal transport.

Optimal transport (OT) theory (Ge et al., 2021; Fatras et al., 2021; Flamary et al., 2021) has become a fundamental tool in various fields, including machine learning (Montesuma et al., 2024; Wang et al., 2024; Yuan et al., 2024) and computer vision (Shi et al., 2024b; 2023). It provides a principled approach for aligning probability distributions and optimizing resource allocation. The OT problem is to minimize the inner product of cost matrix and transport matrix, which is a linear problem. Besides, it is assumed the source and target distributions are fixed. This means OT can not be applied in the size constrained MC problem directly. To address these challenges, we propose the Doubly Bounded Nonlinear Optimal Transport (DB-NOT) problem, which introduces both upper and lower bounds on the transport plan while accommodating non-linear objective functions. This novel formulation

extends the classical OT framework, enabling the modeling of problems with bounded feasibility regions and non-linear optimization goals.

The DB-NOT problem poses significant computational challenges due to the interaction of double-bounded constraints and the complexity of non-linear objective functions. Traditional methods for solving OT problems, such as Sinkhorn iterations (Nguyen et al., 2024) or linear programming (Peyré et al., 2019a), are inadequate in this setting because they are designed for linear or unconstrained formulations. Therefore, there is a pressing need for an optimization algorithm that can efficiently handle DB-NOT problem.

To tackle this problem, we propose the Double-bounded Nonlinear Frank-Wolfe (DNF) method, inspired by the classical Frank-Wolfe algorithm (Jaggi, 2013). The DNF method is specifically tailored to address the challenges of the DB-NOT framework by iteratively optimizing within the feasible region defined by the double-bounded constraints. The core idea of the DNF method is to compute a feasible gradient within the constraint set that best approximates the negative gradient matrix. By searching along this feasible gradient, the DNF method efficiently minimizes the non-linear function while maintaining complicance with the double-bounded constraints. Through iterative updates and convex combinations of feasible gradients, the method ensures that the descent direction remains computationally efficient and effective. To demonstrate the practical utility of our approach, we apply the DNF to size constrained MC clustering. In summary, our contributions are fourfold.

- For the first time, We formulate the Double-bounded Nonlinear Optimal Transport (DB-NOT) problem, which introduces both upper and lower bounds on the transport plan, extending the classical optimal transport framework. This problem seeks to find a transport plan that minimizes a given cost function while satisfying double-bounded constraints, ensuring that the transport plan remains within the prescribed bounds. The DB-NOT problem has applications in diverse fields such as machine learning, economics, and logistics, where practical constraints often require bounded and nonlinear adjustments to classical transport formulations.

- Inspired by the Frank-Wolfe method, we propose the DNF (Double-bounded Nonlinear Frank-Wolfe) method for solving the DB-NOT problem. The DNF method is specifically designed to handle the unique challenges in DB-NOT framework. This approach extends the utility of the Frank-Wolfe method to a broader class of constrained non-linear problems.

- We prove that the DNF method can achieve global optimality regardless of whether the non-linear function is convex or Lipschitz-continuous non-convex. Specifically, the convergence rate is $O(1/t)$ for convex functions and $O(1/\sqrt{t})$ for Lipschitz-continuous non-convex functions. These theoretical results highlight the robustness and versatility of the DNF method across a wide range of problem settings.

- The size constrained min cut clustering framework benefits from the ability of DNF method to handle non-linear constraints effectively, ensuring clusters of appropriate sizes while minimizing the cut value. Experiments on diverse datasets, including image, text, and graph-based data, demonstrate that the DNF-based approach outperforms traditional methods in terms of clustering quality. The results underline the practical applicability and advantages of the proposed method in real-world scenarios.

## 2 PRELIMINARIES

### 2.1 NOTATIONS

The matrices is denoted by tilted capital letters and vectors are presented by lowercase letter. $Z = \{z_1, z_2, \ldots, z_n\} \in \mathbb{R}^{d \times n}$ is the data matrix, in which $d$ is the dimensionality and $n$ is the number of samples. $Y \in Ind \in \mathbb{R}^{n \times c}$ is the indicator matrix, in which each row has only one element of one and the rest elements are zeros. $c$ is the number of clusters. The affinity graph $S$ could be constructed on $Z$ in a number of ways, such as by using Euclidean distance, cosine similarity, or kernel-based methods like the Gaussian kernel function. The Laplacian matrix is $L = D - S$ where $D = diag\{d_{11}, d_{22}, \ldots, d_{nn}\}$ and $d_{ii} = \sum_{j=1}^{n} s_{ij}$. $b_l$ and $b_u$ are the minimum and maximum number of samples in each cluster. The elements in $1_c \in \mathbb{R}^c$ are all ones.

## 2.2 SIZE CONSTRAINTED MIN CUT

Min cut clustering aims to minimize inter-cluster similarities and its objective function is

$$\min_{F \in Ind} Tr(F^T L F) \tag{1}$$

If all samples are assigned to a single cluster, the objective value of problem (1) achieves its minimum of 0. However, such skewed clustering results are typically undesirable. To address this issue, Nie et al. (2024) added doubly bounded constraints to the indicator matrix to prevent the formation of excessively large or overly small clusters. Problem (1) becomes

$$\min_{F \in Ind, b_l 1_c \leq F^T 1_n \leq b_u 1_c} Tr(F^T L F) \iff \min_{F \in Ind, b_l 1_c \leq F^T 1_n \leq b_u 1_c} Tr(F^T (D - S) F)$$

$$\iff \min_{F \in Ind, b_l 1_c \leq F^T 1_n \leq b_u 1_c} 1_n^T S 1_n - Tr(F^T S F) \iff \max_{F \in Ind, b_l 1_c \leq F^T 1_n \leq b_u 1_c} Tr(F^T S F) \tag{2}$$

Nie et al. (2024) solved problem (2) by augmented Lagrangian multiplier method and decoupled the constraints into different variables. However, this introduces additional parameters and variables.

## 3 OUR PROPOSED METHOD

In this section, we will respectively present the double-bounded nonlinear optimal transport perspective of the min cut, the steps of the DNF method, and the basic steps for solving the size constrained min cut using the DNF method.

### 3.1 SIZE CONSTRAINED MIN CUT FROM THE PERSPECTIVE OF DOUBLE-BOUNDED NONLINEAR OPTIMAL TRANSPORT.

we solve the double-bounded problem (2) from the perspective of non-linear optimal transport, which is parameter-free. Since this is an NP-hard problem, $F$ can be relaxed such that the row constraints sum to 1, while the column sums lie within a fixed range. This means that the number of elements in each cluster should fall within an appropriate range, and each element is greater than 0. Specifically, the optimization problem to be solved is given by Eq.(3).

$$\begin{cases} \min_F & J_{MC} = -tr(F^T S F) \\ s.t. & F 1_c = 1_n, b_l 1_c \leq F^T 1_n \leq b_u 1_c, F \geq 0 \end{cases} \tag{3}$$

If we assume the set $\Omega = \{X \mid X 1_c = 1_n, b_l 1_c \leq X^T 1_n \leq b_u 1_c, X \geq 0\}$, then according to the definition, $\Omega$ is called the double-bounded constraint set. The optimization problem can then be simply stated as $\max_{F \in \Omega} J_{MC}$, which is a double-bounded nonlinear optimal transport problem.

Similarly, we can provide an example of a general nonlinear double-bounded optimal transport, which satisfies the following definition.

**Definition 3.1.** Let $\mathcal{H}(F)$ be an arbitrary nonlinear function, and let $\Omega = \{X \mid X 1_c = 1_n, b_l 1_c \leq X^T 1_n \leq b_u 1_c, X \geq 0\}$ be called the double-bounded constraint set. Then, the double-bounded nonlinear optimal transport problem is

$$\min_{F \in \Omega} \mathcal{H}(F) \tag{4}$$

Specifically, if $\mathcal{H}(F)$ is $L$-smooth and convex, then $\min_{F \in \Omega} \mathcal{H}(F)$ is called the $L$-smooth convex double-bounded nonlinear optimal transport problem, abbreviated as $P_{DB}^{L,C}$. If $\mathcal{H}(F)$ is $L$-smooth, then $\min_{F \in \Omega} \mathcal{H}(F) \in P_{DB}^L$.

**Theorem 3.2.** *The size constrained MC problem is a $2\|S\|_F$-smooth double-bounded nonlinear optimal transport problem, i.e., $\max_{F \in \Omega} J_{MC} \in P_{DB}^{2\|S\|_F}$. Proof in A.1.*

For the double-bounded nonlinear optimal transport problem, no existing methods have been able to solve it so far. To address this, we designed a method called Double-bounded Nonlinear Frank-Wolfe (DNF) that can efficiently solve general double-bounded nonlinear optimal transport problems and proved its convergence and convergence rate.

## 3.2 Introduction to the DNF Method.

The core idea of the DNF method is to find a feasible gradient $\partial\mathcal{H}$ within the double-bounded constraint set $\Omega$ that best approximates the negative gradient matrix $-\nabla\mathcal{H}$ of a general nonlinear function $\mathcal{H}$, and to search along the feasible gradient for the optimal value of $\mathcal{H}$ within $\Omega$.

To ensure that the feasible negative gradient $\partial\mathcal{H}$ closely approximates $-\nabla\mathcal{H}$, it is necessary to define a measure $\mathcal{E}(-\nabla\mathcal{H}, \partial\mathcal{H})$ to quantify the degree of approximation. This means solving $\min_{\partial\mathcal{H}\in\Omega}\mathcal{E}(-\nabla\mathcal{H}, \partial\mathcal{H})$. There can be multiple measures for approximation, but not all of them guarantee convergence. Here, we identify two approximation measures: $\mathcal{E}_n(-\nabla\mathcal{H}, \partial\mathcal{H}) = \|\nabla\mathcal{H} + \partial\mathcal{H}\|_F^2$ and $\mathcal{E}_i(-\nabla\mathcal{H}, \partial\mathcal{H}) = \langle\nabla\mathcal{H}, \partial\mathcal{H}\rangle$.

### 3.2.1 The norm-based measure.

Under the norm-based measure, the problem to be solved is $\min_{\partial\mathcal{H}\in\Omega}\mathcal{E}_n(-\nabla\mathcal{H}, \partial\mathcal{H}) = \|\nabla\mathcal{H} + \partial\mathcal{H}\|_F^2$. In practice, $\Omega$ can be viewed as $\Omega = \Omega_1 \cup \Omega_2 \cup \Omega_3$, where $\Omega_1 = \{X \mid X \geq 0, X1_c = 1_n\}$, $\Omega_2 = \{X \mid X^T1_n \geq b_l1_c\}$, $\Omega_3 = \{X \mid X^T1_n \leq b_u1_c\}$.

**Theorem 3.3.** *For $\min_{\partial\mathcal{H}\in\Omega_1}\|\nabla\mathcal{H} + \partial\mathcal{H}\|_F^2$, let $\partial\mathcal{H}_i$ denotes the $i$-th row of $\partial\mathcal{H}$, and $\partial\mathcal{H}_{ij}$ represents the $ij$-th element of $\partial\mathcal{H}$. The optimal solution of $\min_{\partial\mathcal{H}\in\Omega_1}\|\nabla\mathcal{H} + \partial\mathcal{H}\|_F^2$, i.e., the projection onto $\Omega_1$, is given by:*

$$Proj_{\Omega_1}(-\nabla\mathcal{H})_{ij} = \partial\mathcal{H}_{ij}^* = \left((-\nabla\mathcal{H})_{ij} + \eta_i\right)_+ \tag{5}$$

*where $(\cdot)_+$ denotes the positive part, and $\eta$ is determined by $\sum_{j=1}^c \partial\mathcal{H}_{ij}^* = 1$. Proof in A.2.*

For $\min_{\partial\mathcal{H}\in\Omega_2}\|\nabla\mathcal{H} + \partial\mathcal{H}\|_F^2$ or $\min_{\partial\mathcal{H}\in\Omega_3}\|\nabla\mathcal{H} + \partial\mathcal{H}\|_F^2$, the solution can be obtained using a similar projection method.

**Theorem 3.4.** *Assuming $\nabla\mathcal{H}^j$ represents the $j$-th column of $\nabla\mathcal{H}$, the projection of $\min_{\partial\mathcal{H}\in\Omega_2}\|\nabla\mathcal{H} + \partial\mathcal{H}\|_F^2$ onto $\Omega_2$ satisfies Eq.(6). Proof in A.3.*

$$Proj_{\Omega_2}(-\nabla\mathcal{H}^j) = \partial\mathcal{H}^{j*} = \begin{cases} -\nabla\mathcal{H}^j, & if\ (-\nabla\mathcal{H}^j)^T1_n \geq b_l \\ \frac{1}{n}(b_l + 1_n^T\nabla\mathcal{H}^j)1_n - \nabla\mathcal{H}^j, & if\ (-\nabla\mathcal{H}^j)^T1_n < b_l \end{cases} \tag{6}$$

The case of $\Omega_3$ is completely analogous to that of $\Omega_2$. Under the measure $\mathcal{E}_n(-\nabla\mathcal{H}, \partial\mathcal{H}) = \|\nabla\mathcal{H} + \partial\mathcal{H}\|_F^2$, the feasible negative gradient can be found through continuous iterative projection. Specifically, this involves cyclically performing $Proj_{\Omega_1}(-\nabla\mathcal{H})$, $Proj_{\Omega_2}(-\nabla\mathcal{H})$, and $Proj_{\Omega_3}(-\nabla\mathcal{H})$. Since the subsets $\Omega_1$, $\Omega_2$, and $\Omega_3$ are simple sets, by the Dykstra's projection theorem, it can be proven that this iterative procedure will find the optimal projection result (Yuan et al., 2025).

### 3.2.2 The inner product-based measure

Another option for evaluating the feasible negative gradient $\partial\mathcal{H}$ and the negative gradient $-\nabla\mathcal{H}$ is the inner product measure, which involves solving the problem $\min_{\partial\mathcal{H}\in\Omega}\langle\nabla\mathcal{H}, \partial\mathcal{H}\rangle$. In fact, the approximation problem under the inner product measure can be viewed as a form of double-bounded linear optimal transport. Let $\mathcal{G}$ represent the entropy function, where $\mathcal{G}(\partial\mathcal{H}) = \sum_{i,j} \partial\mathcal{H}_{ij}\log(\partial\mathcal{H}_{ij}) - \sum_{i,j} \partial\mathcal{H}_{ij}$.

By introducing entropy regularization, the original problem can be approximated as $\min_{\partial\mathcal{H}\in\Omega}\langle\nabla\mathcal{H}, \partial\mathcal{H}\rangle - \delta\mathcal{G}(\partial\mathcal{H})$, where $\delta > 0$ is the regularization parameter. The approximate gradient obtained by solving the regularized problem is denoted as $\partial_\delta\mathcal{H}^*$. It holds that $\lim_{\delta\to 0}\partial_\delta\mathcal{H}^* = \partial\mathcal{H}^*$, indicating that the solution to the regularized problem converges to the solution of the original problem as $\delta$ approaches zero.

**Theorem 3.5.** *The optimal solution of the problem $\min_{\partial\mathcal{H}\in\Omega}\langle\nabla\mathcal{H}, \partial\mathcal{H}\rangle - \delta\mathcal{G}(\partial\mathcal{H})$ is given by $\partial_\delta\mathcal{H}^* = \mathrm{diag}(u^*)e^{-\nabla\mathcal{H}/\delta}\mathrm{diag}(v^* \odot w^*)$, where $u^*, v^*,$ and $w^*$ are vectors, $\mathrm{diag}()$ represents the operation of creating a diagonal matrix, and $\odot$ denotes the Hadamard (element-wise) product. The vectors $u^*, v^*,$ and $w^*$ can be computed iteratively to convergence using the following update rules:*

$$\begin{cases} u^{(k+1)} = 1_n./(e^{-\nabla\mathcal{H}/\delta}(v^{(k)} \odot w^{(k)})) \\ v^{(k+1)} = \max(b_l1_c./(((e^{-\nabla\mathcal{H}/\delta})^T u^{(k+1)}) \odot w^{(k)}), 1_c) \\ w^{(k+1)} = \min(b_u1_c./(((e^{-\nabla\mathcal{H}/\delta})^T u^{(k+1)}) \odot v^{(k+1)}), 1_c) \end{cases} \tag{7}$$

*where $1./$ denotes element-wise division, $b_l$ and $b_u$ are lower and upper bounds, and $1_n$ and $1_c$ are vectors of ones with appropriate dimensions. Proof in A.4.*

This theorem provides a method for approximating the true negative gradient $-\nabla\mathcal{H}$ using the feasible $\delta$-gradient $\partial_\delta\mathcal{H}^*$ under the inner product measure. The approximation relationship is given by (Bonneel & Digne, 2023):

$$\lim_{\delta\to 0}\partial_\delta\mathcal{H}^* = \partial\mathcal{H}^* = \underset{\partial\mathcal{H}\in\Omega}{\text{argmin}}\,(\mathcal{E}_i(-\nabla\mathcal{H},\partial\mathcal{H})) \tag{8}$$

By deriving feasible gradient methods under different approximation measures, the update mechanism for DNF can be further obtained.

### 3.2.3 PERFORMING OPTIMAL VALUE SEARCH.

In the previous section, we addressed feasible gradient approximation methods under different measures, i.e., $\min_{\partial\mathcal{H}\in\Omega}\mathcal{E}(-\nabla\mathcal{H},\partial\mathcal{H})$. The next step is to perform the search. We choose the $t$-th step size $\mu^{(t)}\in(0,1)$ and update $F^{(t)}$ as follows:

$$F^{(t+1)} \leftarrow (1-\mu^{(t)})F^{(t)} + \mu^{(t)}\partial\mathcal{H}^{*(t)} \tag{9}$$

**Theorem 3.6.** *By arbitrarily choosing $\mu^{(t)}\in(0,1)$, if $F^{(t)}$ satisfies $F^{(t)}\in\Omega$, the updated $F^{(t+1)}$ obtained from the search will also satisfy $F^{(t+1)}\in\Omega$. Proof in A.5.*

Here, we provide three different choices for the search step size and offer convergence proofs for each of these step sizes. To introduce the specific significance of the three step sizes, we introduce the concept of the dual gap. Moreover, we will later demonstrate that the dual gap is an important metric for measuring convergence. Specifically, when the dual gap equals 0, the algorithm reaches either the global optimum or a critical point.

**Definition 3.7.** Define the function $g(F) = \min_{\partial\mathcal{H}\in\Omega}\mathcal{E}(\partial\mathcal{H} - F, \nabla\mathcal{H})$ with respect to $F$. Then, $g(F)$ is called the dual gap function of $\mathcal{H}$. For the inner product measure, the corresponding formula is $g^{(t)} = g(F^{(t)}) = <F^{(t)} - \partial\mathcal{H}^{*(t)}, \nabla\mathcal{H}^{(t)}>$.

**Definition 3.8.** We define three types of step size: the easy step size $\mu_e$, the line search step size $\mu_l$, and the dual step size $\mu_g$. The expressions for the three step sizes are as follows:

$$\begin{cases} \mu_e^{(t)} = \dfrac{2}{t+2} \\[2mm] \mu_l^{(t)} = \underset{\mu\in(0,1)}{\text{argmin}}\,\mathcal{H}\left((1-\mu)F^{(t)} + \mu\partial\mathcal{H}^{*(t)}\right) \\[2mm] \mu_g^{(t)} = \min\left(\dfrac{g(F^{(t)})}{L||\partial\mathcal{H}^{*(t)} - F^{(t)}||_F}, 1\right) \end{cases} \tag{10}$$

In general, we assume that $\mu^{(t)}$ is a step size chosen arbitrarily from the three types mentioned above. Using the inner product measure $\mathcal{E}_i(\partial\mathcal{H}, -\nabla\mathcal{H}) = \langle\nabla\mathcal{H},\partial\mathcal{H}\rangle$ as an example, we provide proofs for two convergence theorems. For convex and Lipschitz-smooth functions, the global optimum can be achieved with a convergence rate of $\mathcal{O}(1/t)$. For non-convex and Lipschitz-smooth functions, in the best-case scenario, the convergence to a critical point occurs at a rate of $\mathcal{O}(1/\sqrt{t})$. We

**Theorem 3.9.** *Assume that $\min_{F\in\Omega}\mathcal{H}\in P_{DB}^{L,C}$ and that $\mathcal{H}$ has a global minimum $F^*$. Then, for any of the step sizes in $\{\mu_e^{(t)}, \mu_l^{(t)}, \mu_g^{(t)}\}$, the following inequality holds:*

$$\mathcal{H}(F^{(t)}) - \mathcal{H}(F^*) \le \frac{4L}{t+1} \tag{11}$$

*Proof in A.6*

**Theorem 3.10.** *Assume that $\min_{F\in\Omega}\mathcal{H}\in P_{DB}^{L}$ and that $\mathcal{H}$ has a local minimum $F^*$. $\tilde{g}^{(t)}$ represents the smallest dual gap $g^{(t)}$ obtained during the first $t$ iterations of the DNF algorithm, i.e., $\tilde{g}^{(t)} = \min_{1\le k\le t} g^{(k)}$. By using $\mu_g^{(t)}$ as step. Then $\tilde{g}^{(t)}$ satisfies the following inequality:*

$$\tilde{g}^{(t)} \le \frac{\max\{2(\mathcal{H}(F^{(0)}) - \mathcal{H}(F^*)), 2nL\}}{\sqrt{t+1}} \tag{12}$$

*Proof in A.7.*

$\tilde{g}(F)$ or $g(F)$ can be used as a criterion for the convergence of the algorithm, due to the following theorem, which states that when $g(F)$ approaches 0, $\mathcal{H}(F^{(t)}) \to \mathcal{H}(F^*)$.

**Theorem 3.11.** *For $F^{(t)} \in \Omega$ and convex function $\mathcal{H}$, $g(F^{(t)}) \geq \mathcal{H}(F^{(t)}) - \min_{F \in \Omega} \mathcal{H}(F) = \mathcal{H}(F^{(t)}) - \mathcal{H}(F^*)$, and when $g^{(t)}$ converges to 0 at $\mathcal{O}(\frac{1}{t})$, it means that $\mathcal{H}(F^{(t)}) - \min_{F \in \Omega} \mathcal{H}(F) = \mathcal{H}(F^{(t)}) - \mathcal{H}(F^*) \to 0$ at $\mathcal{O}(\frac{1}{t})$. More generally, if $\mathcal{H}$ is not a convex function, then $g(F^{(t)}) = 0$ if and only if $F^{(t)}$ is a stable critical point of $\mathcal{H}$. Proof in A.8.*

It is worth noting that Eq.(11) and Eq.(12) provide two completely different conclusions. Eq.(11) applies under the condition of convexity and $L$-smoothness, indicating that when $t$ is sufficiently large, $\mathcal{H}(F^{(t)})$ will converge to $\mathcal{H}(F^*)$ with a convergence rate of $\mathcal{O}(1/t)$. In contrast, Eq.(12) requires only $L$-smoothness, which shows that after enough iterations of the DNF algorithm, the best step will converge to a stable critical point $\mathcal{O}(1/\sqrt{t})$.

### 3.3 DNF METHOD FOR SIZE CONSTRAINED MIN CUT.

For size constrained min cut, it is also modeled as a double-bounded nonlinear optimal transport problem, and is applicable to the DNF method. Specifically, size constrained min cut belongs to $P_{DB}^{2||S||_F}$. For size constrained min cut, where $\mathcal{H} = -\text{tr}(F^T SF)$, we have $\nabla \mathcal{H} = -2SF$. By selecting a search step size $\mu^{(t)}$ and updating according to Eq.(9), we can obtain $\partial \mathcal{H}^{*(t)}$ under a certain measure. Based on the previous theorems, it is easy to derive the following corollary:

**Theorem 3.12.** *By solving the minimum cut problem using the DNF algorithm, after $t$ steps, the best step within $t$ steps always converges to the optimal solution, which satisfies:*

$$\tilde{g}^{(t)} = \min_{1 \leq k \leq t} g^{(k)} \leq \frac{\max\{2(\mathcal{H}(F^{(0)}) - \mathcal{H}(F^*)), 4n\|S\|_F\}}{\sqrt{t+1}}. \tag{13}$$

For the choose of step size about size constrained min cut problem, we have:

**Theorem 3.13.** *For size constrained min cut, its line search step size $\mu_l^{(t)}$ has an analytical solution $\mu_l^{*(t)}$. The specific proof and selection method can be found in A.9.*

Further, we provide the algorithmic process for solving general bilateral nonlinear optimal transport problems using the DNF method, as well as the process for solving the size constrained min cut problem.

---

**Algorithm 1:** Solution for problem (4).

---
  **Input** $\mathcal{H}$
  Initialize the variable
  **repeat**
      Compute $\nabla \mathcal{H}^{(t)}$
      Compute $\partial \mathcal{H}^{*(t)} = argmin\ \mathcal{E}(\partial \mathcal{H}, -\nabla \mathcal{H}^{(t)})$ by Eq.(7) or Theorem3.3 and Theorem3.4
      Updating $F^{(t+1)} \leftarrow (1 - \mu^{(t)})F^{(t)} + \mu^{(t)}\partial \mathcal{H}^{*(t)}$
      Updating the step size $\mu^{(t+1)}$
  **until** convergence
  **Output** the optimal solution

---

The DNF algorithm can be applied to the min cut problem very easily. We simply need to compute the gradient $\nabla \mathcal{H}^{(t)}$ and plug it in. The gradient is $-2SF^{(t)}$. For the calculation of the feasible gradient $\partial \mathcal{H}^{*(t)} = \min_{\partial \mathcal{H} \in \Omega} \mathcal{E}(-\nabla \mathcal{H}^{(t)}, \partial \mathcal{H})$, both the norm measure and the inner product measure can be used. In the following proof, we will use the inner product measure for the demonstration. Specifically,

the computation of the size constrained min cut problem with the DNF algorithm is as follows:

---

**Algorithm 2:** DNF for size constrained min cut problem (3).

---

**Input** $S$
Initialize indicator matrix $F$.
**repeat**
        Compute $\nabla \mathcal{H}^{(t)} = -2SF^{(t)}$
        Compute $\partial \mathcal{H}^{*(t)} = argmin\ \mathcal{E}(\partial \mathcal{H}, -\nabla \mathcal{H}^{(t)})$ by Eq.(7) or Theorem3.3 and Theorem3.4
        Updating the $\mu^{(t)}$ by Theorem3.13 or Eq.(10)
        Updating $F^{(t+1)} \leftarrow (1 - \mu^{(t)})F^{(t)} + \mu^{(t)}\partial \mathcal{H}^{*(t)}$
**until** convergence
**Output** the optimal indicator matrix $F^*$

---

In addition, the idea of DNF can be applied not only to double-bounded Nonlinear Optimal Transport(DB-NOT) but also as a method for other types of nonlinear optimal transport problems.

## 4 TIME COMPLEXITY ANALYSIS

Typically, the similarity matrix $S$ is relatively sparse. Assume that for size constrained min cut, $S \in \mathbb{R}^{n \times n}$, and each row of $S$ contains only $m$ non-zero elements. Then, the time complexity for computing $\nabla \mathcal{H} = -2SF$ is $\mathcal{O}(nmc)$, where $c$ is the number of categories.

In solving the feasible gradient problem, i.e., $\min_{\partial \mathcal{H} \in \Omega} \mathcal{E}(-\nabla \mathcal{H}, \partial \mathcal{H})$, whether solving the norm-based measure problem $\min_{\partial \mathcal{H} \in \Omega} \mathcal{E}(-\nabla \mathcal{H}, \partial \mathcal{H}) = \|\nabla \mathcal{H} + \partial \mathcal{H}\|_F$ or the inner-product-based measure problem $\min_{\partial \mathcal{H} \in \Omega} \langle \partial \mathcal{H}, \nabla \mathcal{H} \rangle$, the core lies in computing the matrix-vector multiplication or element-wise division for inner products. Thus, the time complexity remains $\mathcal{O}(nmc)$. Similarly, the time complexity for updating $F$ is $\mathcal{O}(nc)$, while the minimal update cost for $\mu$ is only $\mathcal{O}(1)$.

In summary, the overall time complexity of our algorithm is $\mathcal{O}(n(m + 1)c)$.

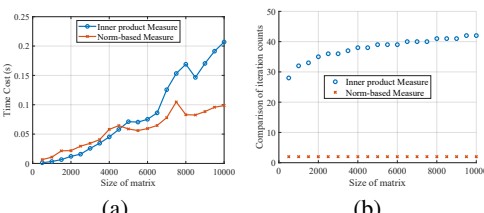

(a)            (b)

Figure 1: Comparison of inner product and norm-based measure in gradient approximation.

## 5 EXPERIMENTS

We evaluated our algorithm on eight real-world datasets, comparing it with twelve comparative methods. All experiments are implemented on a machine with a 3.59GHz R7-3700X processor and 64GB main memory. The analysis covers clustering performance, solution distribution, parameter sensitivity, and convergence, highlighting the robustness and stability of the algorithm. Additional results are shown in Appendix C.

### 5.1 CLUSTERING RESULTS

**Datasets & Baseline** We conducted experiments on eight diverse benchmark datasets (COIL20, Digit, JAFFE, MSRA25, PalmData25, USPS20, Waveform21, MnistData05), covering images, handwriting, and waveforms, with all features normalized to zero mean and unit variance. We compared classic partition-based methods (K-Means, Coordinate Descent K-Means, BKNC (Chen et al., 2022)), graph-based min-cut approaches with normalization and balance regularization (ratio-cut, normalized-cut, FCFC (Liu et al., 2018), Scut (Nie et al., 2024)), and acceleration techniques for spectral clustering (Nystrom (Chen et al., 2011), FSC (Zhu et al., 2017), LSCR (Chen & Cai, 2011), LSCK).

Table 1: Mean clustering performance (%) of compared methods on real-world datasets.

| Metric | Method | COIL20 | Digit | JAFFE | MSRA25 | PalmData25 | USPS20 | Waveform21 | MnistData05 |
|--------|--------|--------|-------|-------|--------|-----------|--------|-----------|------------|
| ACC | KM | 53.44 | 58.33 | 72.16 | 49.33 | 70.32 | 55.51 | 50.38 | 53.86 |
| | CDKM | 52.47 | 65.82 | 80.85 | 59.63 | 76.05 | 57.68 | 50.36 | 54.24 |
| | Rcut | 78.14 | 74.62 | 84.51 | 56.84 | 87.03 | 57.83 | 51.93 | 62.80 |
| | Ncut | 78.88 | 76.71 | 83.76 | 56.23 | 86.76 | 59.20 | 51.93 | 61.14 |
| | Nystrom | 51.56 | 72.08 | 75.77 | 52.85 | 76.81 | 62.55 | 51.49 | 55.91 |
| | BKNC | 57.11 | 60.92 | 93.76 | **65.47** | 86.74 | 62.76 | 51.51 | 52.00 |
| | FCFC | 59.34 | 43.94 | 71.60 | 54.27 | 69.38 | 58.23 | 56.98 | 54.41 |
| | FSC | **82.76** | 79.77 | 81.69 | 56.25 | 82.27 | 67.63 | 50.42 | 57.76 |
| | LSCR | 65.67 | 78.14 | 91.97 | 53.82 | 58.25 | 63.07 | 56.19 | 57.15 |
| | LSCK | 62.28 | 78.04 | 84.98 | 54.41 | 58.31 | 61.86 | 54.95 | 58.57 |
| | Scut | 80.35 | 81.96 | **96.71** | 57.09 | 93.02 | 73.11 | **57.63** | 66.13 |
| | DNF | 81.22 | **85.09** | 96.71 | 57.20 | **93.18** | **73.35** | **57.63** | **66.52** |
| NMI | KM | 71.43 | 58.20 | 80.93 | 60.10 | 89.40 | 54.57 | 36.77 | 49.57 |
| | CDKM | 71.16 | 63.64 | 87.48 | 63.83 | 91.94 | 55.92 | 36.77 | 49.23 |
| | Rcut | 86.18 | 75.28 | 90.11 | 71.64 | 95.41 | 63.84 | 37.06 | 63.11 |
| | Ncut | 86.32 | 76.78 | 89.87 | 71.50 | 95.26 | 64.46 | 37.06 | **63.22** |
| | Nystrom | 66.11 | 70.13 | 82.53 | 57.77 | 93.09 | 59.00 | 36.95 | 48.53 |
| | BKNC | 69.80 | 59.37 | 92.40 | 69.30 | 95.83 | 57.10 | 36.94 | 44.56 |
| | FCFC | 74.05 | 38.33 | 80.30 | 63.34 | 89.47 | 55.71 | 22.89 | 48.75 |
| | FSC | **91.45** | 80.98 | 90.43 | 70.60 | 94.62 | **74.75** | 36.76 | 58.33 |
| | LSCR | 74.67 | 75.07 | 93.13 | 68.06 | 81.84 | 62.36 | 33.37 | 52.82 |
| | LSCK | 74.02 | 76.53 | 87.89 | 67.97 | 81.70 | 65.23 | 36.92 | 59.14 |
| | Scut | 86.23 | 80.63 | **96.24** | 72.61 | 97.47 | 70.89 | 37.65 | 59.84 |
| | DNF | 86.75 | **83.45** | 96.24 | **73.08** | **97.70** | 71.50 | **37.70** | 60.69 |
| ARI | KM | 50.81 | 45.80 | 66.83 | 34.66 | 65.06 | 43.57 | 25.56 | 37.18 |
| | CDKM | 48.11 | 52.74 | 76.36 | 37.70 | 71.73 | 45.59 | 25.56 | 36.79 |
| | Rcut | 73.73 | 65.81 | 81.70 | 46.35 | 84.76 | 51.99 | 25.31 | **51.32** |
| | Ncut | 74.30 | 68.21 | 81.30 | 45.90 | 84.25 | 52.72 | 25.31 | 50.51 |
| | Nystrom | 45.96 | 59.50 | 69.85 | 38.07 | 76.23 | 50.01 | 25.03 | 38.21 |
| | BKNC | 49.96 | 48.98 | 87.96 | 54.78 | 85.56 | 48.43 | 25.02 | 32.89 |
| | FCFC | 54.41 | 25.50 | 65.73 | 40.42 | 66.03 | 46.32 | 22.89 | 36.86 |
| | FSC | **79.46** | 73.03 | 80.26 | 43.99 | 79.67 | **61.71** | 25.10 | 44.78 |
| | LSCR | 57.68 | 67.21 | 86.76 | 43.31 | 48.70 | 52.64 | 25.12 | 41.46 |
| | LSCK | 54.59 | 68.70 | 77.37 | 42.18 | 48.58 | 52.54 | 26.47 | 46.48 |
| | Scut | 75.48 | 73.38 | **93.32** | 48.99 | 91.75 | 60.94 | **27.10** | 50.41 |
| | DNF | 76.21 | **77.87** | 93.32 | **49.33** | **92.13** | 61.46 | **27.10** | 51.11 |

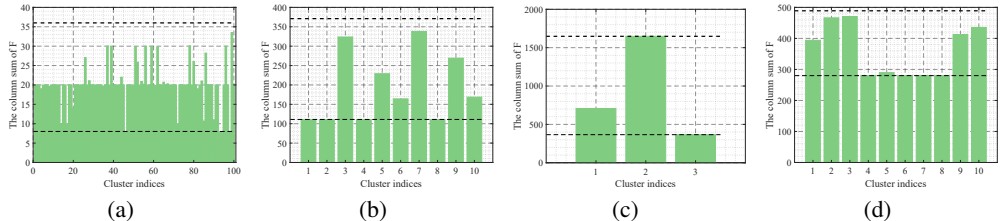

| (a) | (b) | (c) | (d) |

Figure 2: The clustering distribution with lower and upper bounds. (a) PalmData25. (b) USPS20. (c) Waveform21. (d) MnistData05.

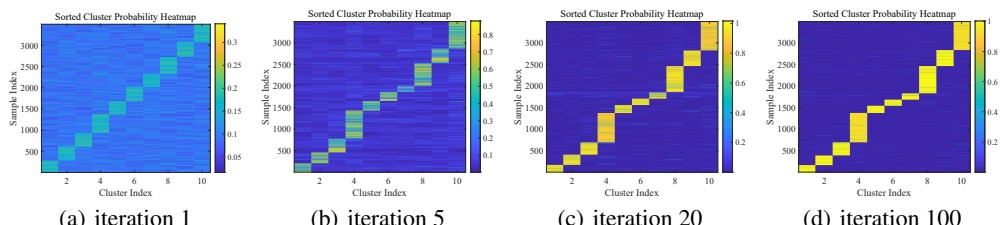

| (a) iteration 1 | (b) iteration 5 | (c) iteration 20 | (d) iteration 100 |

Figure 3: Change of distribution of element values in indicator matrix during the iteration process for MnistData05 dataset.

**Metric & Configuration** Three metrics are applied to comprehensively measure the performance of compared algorithms and proposed method, which are clustering accuracy (ACC), normalized mutual information (NMI) and adjusted rand index (ARI). Larger values of these metrics indicate better clustering performance. In size constrained MC, the affinity graph is constructed by $k$-nn Gaussian kernel function and we adopt the inner product measure to approximate gradient. For simplicity, the bandwidth in Gaussian kernel function is set as the mean Euclidean distances in each dataset and we

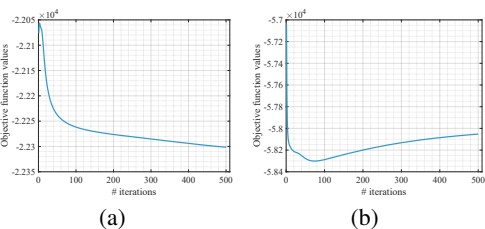

(a)          (b)

Figure 4: Variation of objective function values. (a) PalmData25. (b) MnistData05.

only search the best $k$ in range of $[6, 8, \ldots, 16]$. The number of clusters is set as the true value. Since DNF is gradient-based method, a better initialization is beneficial for the final results. We apply the method in (Nie et al., 2024) to initialize the label matrix. The learning rate is set as easy step size. Ten independent runs are conducted to avoid randomness and the average results are recorded.

**Comparison Results** Table 1 summarizes the clustering performing of various methods across eight real-world datasets. DNF achieves the highest ACC scores on most datasets, particularly excelling on JAFFE, MSRA25 and PalmData25. Our proposed method demonstrates consistent superiority or parity with the top-performing methods on most datasets. Overall, DNF showcases its versatility and effectiveness across diverse datasets, making it a robust choice.

## 5.2 DISCUSSION OF DBNOT

**Approximation of Gradient** In Section 4.2, two measurements are proposed to approximate the gradient within the feasible set: norm-based and inner product-based methods. We compare the running time and number of iterations of these two measures under different matrix sizes, where both methods had the same convergence condition: the change in the optimization variables was less than $10^{-6}$. The experimental results show that when the matrix size is smaller than 4000, the inner product measure consumes less time than the norm-based method. However, as the matrix size increases, the norm-based method outperforms the inner product measure in terms of running time. Additionally, the number of iterations shows that the norm-based method converges in one step, while the inner product method requires progressively more iterations as the matrix size increases, which is also the main factor contributing to the increase in running time. Therefore, it is recommended to use the norm-based measure when dealing with large-scale datasets.

**Clustering Distribution** Two analyses of the resulting indicator matrix are conducted to evaluate the obtained clustering distribution. The first examines whether the column sums of the matrix fall within the feasible region. We visualize the column sums of label matrix in Figure 2, where the black dashed lines represent the lower and upper bounds. It can be observed that all column sums of $F$ lie within the specified range, ensuring that each cluster in the clustering result is meaningful. The second analysis focuses on whether the values of the indicator matrix approach solutions with a clear structure. Figure 3 illustrates how the element values of $F$ evolve over iterations. It is evident that these values gradually shift from being relatively close to approaching 0 or 1, reflecting an distinct clustering structure. This shows our algorithm effectively approximates results similar to those under discrete constraints. **Converge Analysis** Figure 4 presents the convergence curve of the algorithm over 500 iterations. The objective function value is gradually decreasing as the number of iterations increases. It is noted that the objective function does not necessarily decrease monotonically with iterations. Instead, at some iterative point, it will get closest to the critical point.

## 6 CONCLUSION

This paper introduced the Double-bounded Nonlinear Optimal Transport (DB-NOT) framework, which extends classical optimal transport by incorporating upper and lower bounds on the transport plan. To solve this, we proposed the Double-bounded Nonlinear Frank-Wolfe (DNF) method, achieving global optimality for both convex and Lipschitz-smoothness non-convex functions with proven convergence rates. The effectiveness of the DNF method was further demonstrated in a size constrained min cut clustering framework, where it achieved superior performance on diverse datasets. In the future, further work could focus on improving the computational efficiency of the DNF method for large-scale problems and exploring its applications in more tasks.

## 7 STATEMENT

For the reproducibility of this paper, we have submitted the complete anonymized code with fixed random seeds, as detailed in Appendix B.3. In addition, large language models (LLMs) were only used for language polishing.

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

APPENDIX

The appendix is organized in three sections.

## A PROOFS

### A.1 PROOF FOR THEOREM 4.2.

The size constrained min cut problem is a $2\|S\|_F$-smooth double-bounded nonlinear optimal transport problem, i.e., $\min_{F \in \Omega} J_{\text{MC}} \in P_{DB}^{2\|S\|_F}$.

**Lemma A.1.** *For a differentiable function $f$, we say it is L-smooth if $f$ satisfies $\|\nabla^2 f(x)\| \leq L$. Furthermore, $\|\nabla^2 f(x)\| \leq L$ is equivalent to $\forall x, y \in \text{dom}(f)$, $\|\nabla f(y) - \nabla f(x)\| \leq L\|x - y\|$.(Beliakov, 2007)*

**Lemma A.2.** *For any $A \in \mathbb{R}^{n \times n} \in$, $B \in \mathbb{R}^{n \times c}$, we have $\|AB\|_F \leq \|A\|_F \|B\|_F$.*

*Proof.* Now, we prove Lemma A.2. For any $A \in \mathbb{R}^{n \times n} \in$, $B \in \mathbb{R}^{n \times c}$, we have:

$$\|A\|_F = \sqrt{\sum_{i,j} a_{ij}^2}, \quad \|B\|_F = \sqrt{\sum_{i,j} b_{ij}^2}, \quad \|AB\|_F = \sqrt{\sum_{i,j} \left( \sum_s a_{is} b_{sj} \right)^2} \tag{14}$$

Expanding the $\|AB\|_F$ norm gives the following expression.

$$\|AB\|_F = \sqrt{\sum_{i,j} \left( \sum_s a_{is} b_{sj} \right)^2} \leq \sqrt{\sum_{i,j} ((\sum_s a_{is}^2)(\sum_s b_{sj}^2))} = \sqrt{(\sum_i \sum_s a_{is}^2)(\sum_j \sum_s b_{sj}^2)} = \|A\|_F \|B\|_F \tag{15}$$

The first inequality is obtained by the Cauchy-Schwarz inequality, and the second equality is obtained by the rearrangement theorem.

**Theorem A.3.** *The size constrained min cut problem is a $2\|S\|_F$-smooth double-bounded nonlinear optimal transport problem, i.e., $\min_{F \in \Omega} J_{MC} \in P_{DB}^{2\|S\|_F}$*

*Proof.* For $\min_{F \in \Omega} J_{\text{MC}} = -\text{tr}(F^T S F)$, the gradient is $\nabla \mathcal{H} = \nabla J_{\text{MC}} = -2SF$. For any $F_1, F_2 \in \Omega$, we have

$$\|\nabla \mathcal{H}(F_1) - \nabla \mathcal{H}(F_2)\|_F = \|2S(F_1 - F_2)\| \leq 2\|S\|_F \|F_1 - F_2\|_F \tag{16}$$

According to the definition of L-smoothness, $J_{\text{MC}}$ is L-smooth. This means that $\min_{F \in \Omega} J_{\text{MC}} \in P_{DB}^{2\|S\|_F}$.

### A.2 PROOF FOR THEOREM 4.3.

For $\min_{\partial \mathcal{H} \in \Omega_1} \|\nabla \mathcal{H} + \partial \mathcal{H}\|_F$, let $\partial \mathcal{H}_i$ denote the $i$-th row of $\partial \mathcal{H}$, and $\partial \mathcal{H}_{ij}$ represent the $ij$-th element of $\partial \mathcal{H}$. The optimal solution of $\min_{\partial \mathcal{H} \in \Omega_1} \|\nabla \mathcal{H} + \partial \mathcal{H}\|_F$, i.e., the projection onto $\Omega_1$, is given by:

$$\text{Proj}_{\Omega_1}(-\nabla \mathcal{H})_{ij} = \partial \mathcal{H}_{ij}^* = \left( (-\nabla \mathcal{H})_{ij} + \eta_i \right)_+ \tag{17}$$

where $(\cdot)_+$ denotes the positive part, and $\eta$ is determined by the condition $\sum_{j=1}^c \partial \mathcal{H}_{ij}^* = 1$.

*Proof.* Now, we are solving the problem $\min_{\partial \mathcal{H} \in \Omega_1} \| -\nabla \mathcal{H} - \partial \mathcal{H}\|_F = \|\nabla \mathcal{H} + \partial \mathcal{H}\|_F$, where $\Omega_1 = \{X \mid X \geq 0, X 1_c = 1_n\}$. First, write out the Lagrangian function $\mathcal{L}$. Since for $\Omega_1$, the rows are decoupled, we can separately write the Lagrangian function for the $i$-th row.

$$\mathcal{L}(\partial \mathcal{H}_i, \eta, \theta) = \frac{1}{2} \|\nabla \mathcal{H}_i + \partial \mathcal{H}_i\|_F^2 - \eta(\partial \mathcal{H}_i 1_c - 1) - \sum_j \theta_j (\partial \mathcal{H}_{ij}) \tag{18}$$

The necessary conditions for the KKT points can be derived by setting the derivative of the Lagrangian function to zero. Specifically, for the variables $\partial\mathcal{H}_i, \eta$, and $\theta$, we have:

$$\nabla_{(\partial\mathcal{H}_i)}\mathcal{L} = \nabla\mathcal{H}_i + \partial\mathcal{H}_i - \eta 1_c - \theta = 0 \tag{19}$$

This condition ensures that the solution satisfies the KKT conditions (Dutta et al., 2013) for the optimization problem. Further, we obtain:

$$\partial\mathcal{H}_i = -\nabla\mathcal{H}_i + \eta 1_c + \theta \tag{20}$$

Since the constraint is $\theta \geq 0$, we can rearrange and obtain the solution as:

$$\partial\mathcal{H}_{ij}^* = \left((-\nabla\mathcal{H})_{ij} + \eta_i\right)_+ \tag{21}$$

Here, since we are solving for each row $i$, the multiplier $\eta_i$ will be different for each row. To solve for $\eta_i$, we use the constraint $\sum_j \partial\mathcal{H}_{ij} = 1$, i.e., solving the equation $l(\eta) = \left(\sum_j (-\nabla\mathcal{H})_{ij} + \eta_i\right)_+ - 1$ for its root.

### A.3 PROOF FOR THEOREM 4.4.

Assuming $\nabla\mathcal{H}^j$ represents the $j$-th column of $\nabla\mathcal{H}$, the projection of $\min_{\partial\mathcal{H}\in\Omega_2} \|\nabla\mathcal{H} + \partial\mathcal{H}\|_F$ onto $\Omega_2$ satisfies Eq.(22).

$$\text{Proj}_{\Omega_2}(-\nabla\mathcal{H}^j) = \partial\mathcal{H}^{j*} = \begin{cases} -\nabla\mathcal{H}^j, & \text{if } (-\nabla\mathcal{H}^j)^T 1_n \geq b_l \\ \frac{1}{n}(b_l + 1_n^T \nabla\mathcal{H}^j)1_n - \nabla\mathcal{H}^j, & \text{if } (-\nabla\mathcal{H}^j)^T 1_n < b_l \end{cases} \tag{22}$$

*Proof.* First, consider the simple case for the problem, $\min_{\partial\mathcal{H}\in\Omega_2} \|\nabla\mathcal{H} + \partial\mathcal{H}\|_F$ where $\Omega_2 = \{X \mid X^T 1_n \geq b_l 1_c\}$. If $\nabla\mathcal{H}$ itself satisfies $\nabla\mathcal{H} \in \Omega_2$, then no projection is required. In this case, we have:

$$\text{Proj}_{\Omega_2}(-\nabla\mathcal{H}^j) = \partial\mathcal{H}^{j*} = -\nabla\mathcal{H}^j \tag{23}$$

This means that the first row clearly holds.

For the second row, For the second case, the Lagrangian function $\mathcal{L}$ is written as:

$$\mathcal{L}(\partial\mathcal{H}^j, \lambda) = \frac{1}{2}\|\nabla\mathcal{H}^j + \partial\mathcal{H}^j\|_F^2 - \lambda\left((\partial\mathcal{H}^j)^T 1_n - b_l\right) \tag{24}$$

where $\lambda \geq 0$ is the Lagrange multiplier. Considering the gradient of $\mathcal{L}$:

$$\nabla_{(\partial\mathcal{H}^j)}\mathcal{L} = (\partial\mathcal{H}^j + \nabla\mathcal{H}^j) - \lambda 1_n \tag{25}$$

and based on the complementary slackness condition: $\lambda\left(b_l - (\partial\mathcal{H}^j)^T 1_n\right) = 0$. When $\lambda > 0$, it follows that $b_l = (\partial\mathcal{H}^j)^T 1_n$. At this point, $\partial\mathcal{H}^j = \lambda 1_n - \nabla\mathcal{H}^j$. Using the condition $b_l = (\partial\mathcal{H}^j)^T 1_n$, we have $\left(\lambda 1_n - \nabla\mathcal{H}^j\right)^T 1_n = b_l$. Thus, $\lambda = \frac{1}{n}\left(b_l + (\nabla\mathcal{H}^j)^T 1_n\right)$. Substituting this into the expression for $\partial\mathcal{H}^j$, we get:

$$\partial\mathcal{H}^j = \frac{1}{n}\left(b_l + (\nabla\mathcal{H}^j)^T 1_n\right)1_n - \nabla\mathcal{H}^j \tag{26}$$

Using Dykstra's algorithm, we iteratively compute the projections while maintaining correction terms to ensure convergence to the feasible intersection. Specifically, starting with an initial point, we iteratively update:

$$\begin{aligned} \partial\tilde{\mathcal{H}}_1 &= -\nabla\mathcal{H} + z_1, \quad \partial\mathcal{H} \leftarrow \text{Proj}_{\Omega_1}(\partial\tilde{\mathcal{H}}_1), \quad z_1 \leftarrow \partial\tilde{\mathcal{H}}_1 - \partial\mathcal{H}, \\ \partial\tilde{\mathcal{H}}_2 &= \partial\mathcal{H} + z_2, \quad \partial\mathcal{H} \leftarrow \text{Proj}_{\Omega_2}(\partial\tilde{\mathcal{H}}_2), \quad z_2 \leftarrow \partial\tilde{\mathcal{H}}_2 - \partial\mathcal{H}, \\ \partial\tilde{\mathcal{H}}_3 &= \partial\mathcal{H} + z_3, \quad \partial\mathcal{H} \leftarrow \text{Proj}_{\Omega_3}(\partial\tilde{\mathcal{H}}_3), \quad z_3 \leftarrow \partial\tilde{\mathcal{H}}_3 - \partial\mathcal{H}. \end{aligned} \tag{27}$$

These steps are repeated iteratively until convergence, ensuring that $\partial\mathcal{H}$ satisfies all constraints in $\Omega_1 \cap \Omega_2 \cap \Omega_3$. We can solve the feasible gradient computation problem under the norm measure3.(Størmer, 1972; Tibshirani, 2017)

---

**Algorithm 3:** Dykstra's Algorithm for Feasible Gradient Computation

---

1: **Input:** $\nabla\mathcal{H}$, constraints $\Omega_1, \Omega_2, \Omega_3$
2: **Output:** $\partial\mathcal{H}^*$
3: Initialize $\partial\mathcal{H} = -\nabla\mathcal{H}$, dual variables $z_1 = z_2 = z_3 = 0$
4: **while** not converged **do**
5: $\quad \partial\tilde{\mathcal{H}} \leftarrow \partial\mathcal{H} + z_1, \quad \partial\mathcal{H} \leftarrow \text{Proj}_{\Omega_1}(\partial\tilde{\mathcal{H}}), \quad z_1 \leftarrow \partial\tilde{\mathcal{H}} - \partial\mathcal{H}$
6: $\quad \partial\tilde{\mathcal{H}} \leftarrow \partial\mathcal{H} + z_2, \quad \partial\mathcal{H} \leftarrow \text{Proj}_{\Omega_2}(\partial\tilde{\mathcal{H}}), \quad z_2 \leftarrow \partial\tilde{\mathcal{H}} - \partial\mathcal{H}$
7: $\quad \partial\tilde{\mathcal{H}} \leftarrow \partial\mathcal{H} + z_3, \quad \partial\mathcal{H} \leftarrow \text{Proj}_{\Omega_3}(\partial\tilde{\mathcal{H}}), \quad z_3 \leftarrow \partial\tilde{\mathcal{H}} - \partial\mathcal{H}$
8: **end while**
9: **Return:** $\partial\mathcal{H}^*$

---

### A.4 PROOF FOR THEOREM 4.5.

The optimal solution of the problem $\min_{\partial\mathcal{H}\in\Omega}\langle\partial\mathcal{H}, \nabla\mathcal{H}\rangle - \delta\mathcal{G}(\partial\mathcal{H})$ is given by $\partial_\delta\mathcal{H}^* = \text{diag}(u^*)e^{-\nabla\mathcal{H}/\delta}\text{diag}(v^* \odot w^*)$, where $u^*, v^*$, and $w^*$ are vectors, $\text{diag}(\cdot)$ represents the operation of creating a diagonal matrix, and $\odot$ denotes the Hadamard (element-wise) product. The vectors $u^*, v^*$, and $w^*$ can be computed iteratively to convergence using the following update rules:

$$\begin{cases} u^{(k+1)} = 1_n./(e^{-\nabla\mathcal{H}/\delta}(v^{(k)} \odot w^{(k)})) \\ v^{(k+1)} = \max(b_l 1_c./((( e^{-\nabla\mathcal{H}/\delta})^T u^{(k+1)}) \odot w^{(k)}), 1_c) \\ w^{(k+1)} = \min(b_u 1_c./((( e^{-\nabla\mathcal{H}/\delta})^T u^{(k+1)}) \odot v^{(k+1)}), 1_c) \end{cases} \tag{28}$$

where $1_n./$ denotes element-wise division, $b_l$ and $b_u$ are lower and upper bounds, and $1_n$ and $1_c$ are vectors of ones with appropriate dimensions.

*Proof.* The Lagrangian function for solving the feasible gradient problem based on the inner product measure, defined as $\min_{\partial\mathcal{H}\in\Omega}\langle\partial\mathcal{H}, \nabla\mathcal{H}\rangle - \delta\mathcal{G}(\partial\mathcal{H})$, where $\Omega = \{X \mid X1_c = 1_n, b_l 1_c \leq X^T 1_n \leq b_u 1_c, X \geq 0\}$, is written as:

$$\mathcal{L}(\partial\mathcal{H}, \eta, \lambda, \nu) = \langle\partial\mathcal{H}, \nabla\mathcal{H}\rangle - \delta\mathcal{G}(\partial\mathcal{H}) + \eta^T(\partial\mathcal{H}1_c - 1_n) + \lambda^T(b_l 1_c - \partial\mathcal{H}^T 1_n) + \nu^T(\partial\mathcal{H}^T 1_n - b_u 1_c) \tag{29}$$

where $\eta \in \mathbb{R}^n$, $\lambda, \nu \in \mathbb{R}^c_{\geq 0}$ are Lagrange multipliers corresponding to the equality and inequality constraints. Let $\mathcal{L}$ be differentiated with respect to $\partial\mathcal{H}$ and set to zero, i.e.,

$$\nabla_{(\partial\mathcal{H})}\mathcal{L} = \nabla\mathcal{H} - \delta\nabla\mathcal{G}(\partial\mathcal{H}) + \eta 1_c^T - 1_n\lambda^T + 1_n\nu^T = 0 \tag{30}$$

Since $\mathcal{G}(\partial\mathcal{H}) = \sum_{ij}\partial\mathcal{H}_{ij}\log(\partial\mathcal{H}_{ij}) - \sum_{ij}\partial\mathcal{H}_{ij}$, consider the $ij$-th element of $\nabla_{(\partial\mathcal{H})}\mathcal{L}$ and substitute $\mathcal{G}(\partial\mathcal{H})$, which gives:

$$\nabla_{(\partial\mathcal{H})}\mathcal{L}_{ij} = \nabla\mathcal{H}_{ij} + \delta\log(\partial\mathcal{H}_{ij}) + \eta_i - \lambda_j + \nu_j = 0 \tag{31}$$

This implies: $-\nabla\mathcal{H}_{ij} - \eta_i + (\lambda_j - \nu_j) = \delta\log(\partial\mathcal{H}_{ij})$, which leads to:

$$(\partial_\delta\mathcal{H}^*)_{ij} = e^{-\frac{\eta_i}{\delta}}e^{-\frac{\nabla\mathcal{H}_{ij}}{\delta}}e^{\frac{\lambda_j-\nu_j}{\delta}} \tag{32}$$

Since $\lambda \geq 0$ and $\nu \geq 0$, we set

$$\begin{cases} u = e^\eta \\ v = e^\lambda, e^\lambda \geq 1_n \\ w = e^{-\nu}, e^{-\nu} \leq 1_n \end{cases} \tag{33}$$

Further, we can derive the following formula:

$$(\partial_\delta\mathcal{H}^*) = \text{diag}(e^{-\frac{\eta}{\delta}})e^{-\frac{\nabla\mathcal{H}}{\delta}}\text{diag}(e^{\frac{\lambda-\nu}{\delta}}) = \text{diag}(u)e^{-\frac{\nabla\mathcal{H}}{\delta}}\text{diag}(v \odot w) \tag{34}$$

Since we aim to compute $\partial\mathcal{H}^*$ and $\lim_{\delta\to 0}\partial_\delta\mathcal{H}^* = \partial\mathcal{H}^*$, it suggests that $\delta$ should not be taken too large. Thus, the conclusion is:

$$(\partial_\delta\mathcal{H}^*) = \text{diag}(u)e^{-\frac{\nabla\mathcal{H}}{\delta}}\text{diag}(v \odot w) \tag{35}$$

The next step is to derive the iteration formula.

$$
\begin{cases}
u^{(k+1)} = 1_n./(e^{-\nabla\mathcal{H}/\delta}(v^{(k)} \odot w^{(k)})) \\
v^{(k+1)} = \max(b_l 1_c./(((e^{-\nabla\mathcal{H}/\delta})^T u^{(k+1)}) \odot w^{(k)}), 1_c) \\
w^{(k+1)} = \min(b_u 1_c./(((e^{-\nabla\mathcal{H}/\delta})^T u^{(k+1)}) \odot v^{(k+1)}), 1_c)
\end{cases}
\tag{36}
$$

Since $\partial_\delta \mathcal{H}^* 1_c = 1_n$, we can derive that:

$$
\mathrm{diag}(u) e^{-\frac{\nabla\mathcal{H}}{\delta}} \mathrm{diag}(v \odot w) 1_c = \mathrm{diag}(u) e^{-\frac{\nabla\mathcal{H}}{\delta}}(v \odot w) = u \odot \left(e^{-\frac{\nabla\mathcal{H}}{\delta}}(v \odot w)\right) = 1_n
\tag{37}
$$

Based on this, we can derive:

$$
u = 1_n./(e^{-\frac{\nabla\mathcal{H}}{\delta}}(v \odot w)) \Rightarrow u^{(k+1)} = 1_n./\left(e^{-\frac{\nabla\mathcal{H}}{\delta}}(v^{(k)} \odot w^{(k)})\right)
\tag{38}
$$

Here, the $1_n./$ represents element-wise division. At the same time, there is the constraint: $b_l 1_c \leq (\partial_\delta \mathcal{H}^*)^T 1_n \leq b_u 1_c$, which can be expressed as:

$$
b_l 1_c \leq \left(\mathrm{diag}(u) e^{-\frac{\nabla\mathcal{H}}{\delta}} \mathrm{diag}(v \odot w)\right)^T 1_n = v \odot w \odot \left((e^{-\frac{\nabla\mathcal{H}}{\delta}})^T u\right) \leq b_u 1_c
\tag{39}
$$

First, we separately consider the constraint: $b_l 1_c \leq v \odot w \odot \left((e^{-\frac{\nabla\mathcal{H}}{\delta}})^T u\right)$ and based on the complementary slackness condition:

$$
\lambda^T \left(b_l 1_c - v \odot w \odot \left((e^{-\frac{\nabla\mathcal{H}}{\delta}})^T u\right)\right) = 0
\tag{40}
$$

This leads to the following cases for discussion:

$$
\begin{cases}
v \odot w \odot \left((e^{-\frac{\nabla\mathcal{H}}{\delta}})^T u\right) \leq b_l 1_c \Rightarrow v \leq b_l 1_c./((e^{-\frac{\nabla\mathcal{H}}{\delta}})^T u \odot w), & \lambda = 0 \\
v \odot w \odot \left((e^{-\frac{\nabla\mathcal{H}}{\delta}})^T u\right) = b_l 1_c \Rightarrow v = b_l 1_c./((e^{-\frac{\nabla\mathcal{H}}{\delta}})^T u \odot w), & \lambda > 0
\end{cases}
\tag{41}
$$

Given $v = e^\lambda$, based on the definition, when $\lambda = 0$, we have $v = 1_c$. Therefore, the above equation should be updated as:

$$
\begin{cases}
v = 1_c, & \lambda = 0 \\
v = b_l 1_c./((e^{-\frac{\nabla\mathcal{H}}{\delta}})^T u \odot w), & \lambda > 0
\end{cases}
\tag{42}
$$

In summary, the update iteration formula for $v$ can be expressed as:

$$
v^{(k+1)} = \max(b_l./(((e^{-\nabla\mathcal{H}/\delta})^T u^{(k+1)}) \odot w^{(k)}), 1_c)
\tag{43}
$$

Similarly, based on the complementary slackness condition for $w$, its two cases can be derived as:

$$
\begin{cases}
w = 1_c, & \nu = 0, \\
w = b_u 1_c./((e^{-\nabla\mathcal{H}/\delta})^T u \odot v), & \nu > 0.
\end{cases}
\tag{44}
$$

Based on the definition of $w$, $w = e^{-\nu}$. When $\nu > 0$, $w \leq 1_c$. This implies that the update formula for $w$ should be as follows:

$$
w^{(k+1)} = \min(b_u 1_c./(((e^{-\nabla\mathcal{H}/\delta})^T u^{(k+1)}) \odot v^{(k+1)}), 1_c)
\tag{45}
$$

Thus, the update formulas for $u$, $v$, and $w$ can be obtained as follows. Using these formulas, the feasible gradient problem under the inner product measure $\min_{\partial\mathcal{H} \in \Omega} \langle \partial\mathcal{H}, \nabla\mathcal{H} \rangle - \delta\mathcal{G}(\partial\mathcal{H})$ can be effectively solved.

That is, we have derived $\partial_\delta \mathcal{H}^*$, and by selecting a sufficiently small $\delta$, we can obtain a good approximation of the feasible gradient $\partial\mathcal{H}^*$.

## A.5 Proof for Theorem 4.6.

By arbitrarily choosing $\mu^{(t)} \in (0,1)$, if the initial $F^{(t)}$ satisfies $F^{(t)} \in \Omega$, the updated $F^{(t+1)}$ obtained from the search will also satisfy $F^{(t+1)} \in \Omega$, where

$$F^{(t)} \leftarrow (1 - \mu^{(t)})F^{(t)} + \mu^{(t)}\partial\mathcal{H}^{(t)} \tag{46}$$

*Proof.* The proof of this theorem is straightforward. Since $\Omega = \{X \mid X1_c = 1_n, b_l 1_c \leq X^T 1_n \leq b_u 1_c, X \geq 0\}$, we first prove that $\Omega$ is a convex set. For all $X_1, X_2 \in \Omega$ and $\alpha \in (0,1)$, we have:

$$\begin{cases} (\alpha X_1 + (1-\alpha)X_2)1_c = \alpha(X_1 1_c) + (1-\alpha)(X_2 1_c) = \alpha 1_n + (1-\alpha)1_n = 1_n \\ b_l 1_c \leq \alpha(X_1^T 1_n) + (1-\alpha)(X_2^T 1_n) \leq b_u 1_c \\ (\alpha X_1 + (1-\alpha)X_2) \geq 0 \end{cases} \tag{47}$$

Thus, $\alpha X_1 + (1-\alpha)X_2 \in \Omega$. Since $\mu^{(t)} \in (0,1)$, the updated $F^{(t+1)}$ is a convex combination of $F^{(t)}$ and $\partial\mathcal{H}^{*(t)}$.(Marcucci et al., 2024) Specifically, $\partial\mathcal{H}^{*(t)} = \arg\min_{\partial\mathcal{H} \in \Omega} \mathcal{E}(-\nabla\mathcal{H}^{(t)}, \partial\mathcal{H})$, meaning $\partial\mathcal{H}^{*(t)} \in \Omega$. As long as we choose $F^{(1)} \in \Omega$, by induction, we can conclude that $F^{(t+1)} \in \Omega$.

Although the proof is simple, its significance is important because this theorem shows that all our search steps involve convex combinations, and they remain within $\Omega$. This allows us to perform a more daring search, which can help in proposing various methods for selecting learning rates.

## A.6 Proof for Theorem 4.9.

Assume that $min_{F \in \Omega}\mathcal{H} \in P_{DB}^{L,C}$ and that $\mathcal{H}$ has a local minimum $F^*$. Then, for any of the step sizes in $\{\mu_e^{(t)}, \mu_l^{(t)}, \mu_g^{(t)}\}$, the following inequality holds:

$$\mathcal{H}(F^{(t)}) - \mathcal{H}(F^*) \leq \frac{4L}{t+1} \tag{48}$$

**Lemma A.4.** *The first-order necessary and sufficient condition for a differentiable convex function $\mathcal{H}(F)$ is*

$$\mathcal{H}(F^{(1)}) - \mathcal{H}(F^{(2)}) \geq \langle F^{(1)} - F^{(2)}, \nabla\mathcal{H}(F^{(2)})\rangle \tag{49}$$

(Rotaru et al., 2024)

**Lemma A.5.** *For a differentiable function $\mathcal{H}(F)$, we say it is L-smooth if $\mathcal{H}(F)$ satisfies $\|\nabla^2\mathcal{H}(F)\| \leq L$. Furthermore, L-smooth is equivalent to*

$$\mathcal{H}(F^{(1)}) \leq \mathcal{H}(F^{(2)}) + \langle\nabla\mathcal{H}(F^{(2)}), (F^{(1)} - F^{(2)})\rangle + \frac{L}{2}\|F^{(1)} - F^{(2)}\|_F^2 \tag{50}$$

*for all $F^{(1)}$ and $F^{(2)}$. (Liu et al., 2022)*

**Lemma A.6.** *The dual gap is defined as $g^{(t)}(F) = g(F^{(t)}) = \langle F^{(t)} - \partial\mathcal{H}^{*(t)}, \nabla\mathcal{H}^{(t)}\rangle$. For a convex function $\mathcal{H}(F)$, let the global optimum be $F^*$. Then, we have the inequality:*

$$g(F^{(t)}) \geq \mathcal{H}(F^{(t)}) - \mathcal{H}(F^*) \tag{51}$$

*Proof.* Based on the dual gap, we can obtain the following equation:

$$g(F^{(t)}) = \langle F^{(t)} - \partial\mathcal{H}^{*(t)}, \nabla\mathcal{H}^{(t)}\rangle = \langle F^{(t)}, \nabla\mathcal{H}^{(t)}\rangle - \langle\partial\mathcal{H}^{*(t)}, \nabla\mathcal{H}^{(t)}\rangle \tag{52}$$

$$= \langle F^{(t)}, \nabla\mathcal{H}^{(t)}\rangle - min_{\partial\mathcal{H} \in \Omega}\langle\partial\mathcal{H}, \nabla\mathcal{H}^{(t)}\rangle \tag{53}$$

$$\geq \langle F^{(t)}, \nabla\mathcal{H}^{(t)}\rangle - \langle F^*, \nabla\mathcal{H}^{(t)}\rangle \tag{54}$$

$$= \langle F^{(t)} - F^*, \nabla\mathcal{H}^{(t)}\rangle \tag{55}$$

Since $\mathcal{H}(F)$ is a convex function, by the first-order condition of convex functions, we have:

$$\langle F^{(t)} - F^*, \nabla\mathcal{H}^{(t)}\rangle \geq \mathcal{H}(F^{(t)}) - \mathcal{H}(F^*) \tag{56}$$

In conclusion, we have proven that:

$$g(F^{(t)}) \geq \mathcal{H}(F^{(t)}) - \mathcal{H}(F^*) \tag{57}$$

**Lemma A.7.** *For a convex function $\mathcal{H}(F)$, at any optimal point $F^*$, the dual gap satisfies $g(F^*) = 0$.*

*Proof.* For a convex function at the optimal point $F^*$, by definition, the dual gap $g(F^*) = \langle \nabla \mathcal{H}^*, F^* - F \rangle$. Since the first-order condition holds, we have:

$$g(F^*) = \langle \nabla \mathcal{H}^*, F^* - \partial \mathcal{H}^* \rangle \leq \mathcal{H}(F^*) - \mathcal{H}(\partial \mathcal{H}^*) \leq 0 \tag{58}$$

By Lemma $A.6$, we also know that:

$$g(F^*) \geq \mathcal{H}(F^*) - \mathcal{H}(F^*) \geq 0 \tag{59}$$

Therefore, we conclude that $g(F^*) = 0$.

**Theorem A.8.** *Assume that $\min_{F \in \Omega} \mathcal{H} \in P_{DB}^{L,C}$ and that $\mathcal{H}$ has a global minimum $F^*$. Then, for any of the step sizes in $\{\mu_e^{(t)}, \mu_l^{(t)}, \mu_g^{(t)}\}$, the following inequality holds:*

$$\mathcal{H}(F^{(t)}) - \mathcal{H}(F^*) \leq \frac{4L}{t+1} \tag{60}$$

*Proof.* First, since we assumed that $\mathcal{H}(F)$ is L-smooth, by Lemma A.5, we have:

$$\mathcal{H}(F^{(t+1)}) \leq \mathcal{H}(F^{(t)}) + \langle \nabla \mathcal{H}(F^{(t)}), (F^{(t+1)} - F^{(t)}) \rangle + \frac{L}{2} \|F^{(t+1)} - F^{(t)}\|_F^2 \tag{61}$$

This inequality expresses that the value of $\mathcal{H}(F)$ at the next step is bounded by its current value plus a linear term and a quadratic term involving the smoothness constant $L$. The update strategy is given by:

$$F^{(t+1)} = (1 - \mu^{(t)}) F^{(t)} + \mu^{(t)} \partial \mathcal{H}^{*(t)} \tag{62}$$

According to the definition of the dual gap,

$$g^{(t)}(F) = g(F^{(t)}) = \langle \partial \mathcal{H}^{*(t)} - F^{(t)}, \nabla \mathcal{H}^{(t)} \rangle \tag{63}$$

this implies that

$$\mathcal{H}(F^{(t+1)}) \leq \mathcal{H}(F^{(t)}) + \langle \nabla \mathcal{H}(F^{(t)}), (F^{(t+1)} - F^{(t)}) \rangle + \frac{L}{2} \|F^{(t+1)} - F^{(t)}\|_F^2 \tag{64}$$

$$= \mathcal{H}(F^{(t)}) + \mu^{(t)} \langle \nabla \mathcal{H}(F^{(t)}), (\partial \mathcal{H}^{*(t)} - F^{(t)}) \rangle + \frac{L}{2} (\mu^{(t)})^2 \|\partial \mathcal{H}^{*(t)} - F^{(t)}\|_F^2 \tag{65}$$

$$= \mathcal{H}(F^{(t)}) - \mu^{(t)} g(F^{(t)}) + \frac{L}{2} (\mu^{(t)})^2 \|\partial \mathcal{H}^{*(t)} - F^{(t)}\|_F^2 \tag{66}$$

At this point, we have provided a bound for $\mathcal{H}(F^{(t+1)})$ and $\mathcal{H}(F^{(t)})$. Assuming that $\mathcal{H}(F^*)$ is the global optimal point within $\Omega$, for the inequality

$$\mathcal{H}(F^{(t+1)}) \leq \mathcal{H}(F^{(t)}) - \mu^{(t)} g(F^{(t)}) + \frac{L}{2} (\mu^{(t)})^2 \|\partial \mathcal{H}^{*(t)} - F^{(t)}\|_F^2 \tag{67}$$

subtracting $\mathcal{H}(F^*)$ from both sides gives the following expression:

$$\mathcal{H}(F^{(t+1)}) - \mathcal{H}(F^*) \leq \mathcal{H}(F^{(t)}) - \mathcal{H}(F^*) - \mu^{(t)} g(F^{(t)}) + \frac{L}{2} (\mu^{(t)})^2 \|\partial \mathcal{H}^{*(t)} - F^{(t)}\|_F^2 \tag{68}$$

Assume $\mathcal{M}(F^{(t)}) = \mathcal{H}(F^{(t)}) - \mathcal{H}(F^*)$, we have:

$$\mathcal{M}(F^{(t+1)}) \leq \mathcal{M}(F^{(t)}) - \mu^{(t)} g(F^{(t)}) + \frac{L}{2} (\mu^{(t)})^2 \|\partial \mathcal{H}^{*(t)} - F^{(t)}\|_F^2 \tag{69}$$

This expression shows how the difference between the objective function $\mathcal{H}(F^{(t)})$ and the global optimum $\mathcal{H}(F^*)$ evolves after the update step, depending on the gradient $g(F^{(t)})$ and the step size $\mu^{(t)}$. Based on the inequality

$$g(F^{(t)}) \geq \mathcal{H}(F^{(t)}) - \mathcal{H}(F^*) \tag{70}$$

the following equation holds:

$$\mathcal{M}(F^{(t+1)}) \leq \mathcal{M}(F^{(t)}) - \mu^{(t)}g(F^{(t)}) + \frac{L}{2}(\mu^{(t)})^2\|\partial\mathcal{H}^{*(t)} - F^{(t)}\|_F^2 \tag{71}$$

$$\leq \mathcal{M}(F^{(t)}) - \mu^{(t)}\mathcal{M}(F^{(t)}) + \frac{L}{2}(\mu^{(t)})^2\|\partial\mathcal{H}^{*(t)} - F^{(t)}\|_F^2 \tag{72}$$

$$= (1 - \mu^{(t)})\mathcal{M}(F^{(t)}) + \frac{L}{2}(\mu^{(t)})^2\|\partial\mathcal{H}^{*(t)} - F^{(t)}\|_F^2 \tag{73}$$

$$\leq (1 - \mu^{(t)})\mathcal{M}(F^{(t)}) + \sup_{\partial\mathcal{H}\in\Omega}\left(\frac{L}{2}(\mu^{(t)})^2\|\partial\mathcal{H} - F^{(t)}\|_F^2\right) \tag{74}$$

$$= (1 - \mu^{(t)})\mathcal{M}(F^{(t)}) + \frac{L}{2}(\mu^{(t)})^2\sup_{\partial\mathcal{H}\in\Omega}\left(\|\partial\mathcal{H} - F^{(t)}\|_F^2\right) \tag{75}$$

$$\leq (1 - \mu^{(t)})\mathcal{M}(F^{(t)}) + \frac{L}{2}(\mu^{(t)})^2\sup_{\forall\partial\mathcal{H},F\in\Omega}\left(\|\partial\mathcal{H} - F\|_F^2\right) \tag{76}$$

The next step is to discuss the value of $\sup_{\forall\partial\mathcal{H},F\in\Omega}\left(\|\partial\mathcal{H} - F\|_F^2\right)$. $\sup_{\partial\mathcal{H}\in\Omega}\left(\|\partial\mathcal{H} - F\|_F^2\right)$ represents the maximum value of the Frobenius norm difference between $\partial\mathcal{H}$ and $F$. To achieve the maximum, $\partial\mathcal{H} - F$ must follow a discrete distribution, ensuring that the positions where $\partial\mathcal{H}$ equals 1 differ from the positions where $F$ equals 1. Consequently, this leads to:

$$\sup_{\partial\mathcal{H}\in\Omega}\left(\|\partial\mathcal{H} - F\|_F^2\right) \leq \sum_i(1^2 + 1^2)n = 2n \tag{77}$$

Substituting the above result, we obtain the following formula:

$$\mathcal{M}(F^{(t+1)}) \leq (1 - \mu^{(t)})\mathcal{M}(F^{(t)}) + (\mu^{(t)})^2Ln \tag{78}$$

Next, we will separately prove that choosing any of the three learning rates satisfies the following inequality:

$$\mathcal{H}(F^{(t)}) - \mathcal{H}(F^*) = \mathcal{M}(F^{(t)}) \leq \frac{4L}{t+1} \tag{79}$$

$\star$ Choose a simple step size $\mu^{(t)} = \mu_e^{(t)} = \frac{2}{t+2}$. Consider the first iteration:

$$\mathcal{M}(F^{(1)}) \leq (1 - \mu^{(0)})\mathcal{M}(F^{(0)}) + (\mu^{(0)})^2Ln \tag{80}$$

where $\mu^{(0)} = \frac{2}{t+2}\big|_{t=1} = \frac{2}{3}$. That is,

$$\mathcal{M}(F^{(1)}) \leq \frac{1}{3}\mathcal{M}(F^{(0)}) + \frac{4}{9}Ln \tag{81}$$

Assume $\mathcal{M}(F^{(0)}) \leq \frac{14}{3}nL$, which is an assumption that can be easily satisfied. Substituting this into the inequality, we have:

$$\mathcal{M}(F^{(1)}) \leq \frac{1}{3}\mathcal{M}(F^{(0)}) + \frac{4}{9}Ln = \frac{14}{9}Ln + \frac{4}{9}Ln = 2Ln = \frac{4nL}{t+1}\big|_{t=1} \tag{82}$$

Using induction, we assume that $\mathcal{M}(F^{(t)}) \leq \frac{4nL}{t+1}$. For the next iteration, we analyze $\mathcal{M}(F^{(t+1)})$ as follows: Using induction, we assume that $\mathcal{M}(F^{(t)}) \leq \frac{4nL}{t+1}$. For the next iteration, we analyze $\mathcal{M}(F^{(t+1)})$ as follows:

$$\mathcal{M}(F^{(t+1)}) \leq \left(1 - \frac{2}{t+2}\right)\mathcal{M}(F^{(t)}) + \left(\frac{2}{t+2}\right)^2nL \tag{83}$$

Substitute the inductive hypothesis $\mathcal{M}(F^{(t)}) \leq \frac{4nL}{t+1}$:

$$\mathcal{M}(F^{(t+1)}) \leq \frac{t}{t+2}\frac{4nL}{t+1} + \left(\frac{2}{t+2}\right)^2nL \tag{84}$$

Simplify the first term:

$$\frac{t}{t+2}\frac{4nL}{t+1} = \frac{4nL}{t+2}\frac{t}{t+1} \tag{85}$$

Combine the two terms:

$$\mathcal{M}(F^{(t+1)}) \leq \frac{4nL}{t+2}\frac{t}{t+1} + \left(\frac{2}{t+2}\right)^2 nL \tag{86}$$

Approximate $\frac{t}{t+1} \leq \frac{t+1}{t+2}$:

$$\mathcal{M}(F^{(t+1)}) \leq \frac{4nL}{t+2}\frac{t+1}{t+2} + \left(\frac{2}{t+2}\right)^2 nL \tag{87}$$

Factorize $\frac{t+1}{t+2}$ in the first term:

$$\mathcal{M}(F^{(t+1)}) \leq \frac{4(t+2)nL}{(t+2)^2} = \frac{4nL}{t+2} \tag{88}$$

Thus, by induction:

$$\mathcal{M}(F^{(t+1)}) \leq \frac{4nL}{t+2} = \frac{4nL}{(t+1)+1} \tag{89}$$

Thus, we have proven that by choosing the simple step size $\mu^{(t)} = \mu_e^{(t)} = \frac{2}{t+2}$, the term $\mathcal{M}(F^{(t)}) = \mathcal{H}(F^{(t)}) - \mathcal{H}(F^*)$ converges at a rate of $\frac{4nL}{t+1}$.

$\star$ Choose the line search step size $\mu_l^{(t)} = \underset{\mu \in (0,1)}{argmin}\ \mathcal{H}(F^{(t)})\left((1-\mu)F^{(t)} + \mu \partial \mathcal{H}(F^{(t)})^{*(t)}\right)$ Assume that at the $t+1$-th update step, we obtain $F^{(t+1)}$, where $F^{(t+1)}$ is derived using a line search step size, while $\tilde{F}^{(t+1)}$ is derived using the aforementioned simple step size. According to the definition, $\mathcal{M}(F^{(t+1)}) \leq \mathcal{M}(\tilde{F}^{(t+1)})$. Furthermore, we can similarly derive the following:

$$\mathcal{M}(F^{(t+1)}) \leq \mathcal{M}(\tilde{F}^{(t+1)}) \leq (1 - \frac{2}{t+2})\mathcal{M}(F^{(t)}) + (\frac{2}{t+2})^2 nL \leq \frac{t}{t+2}\frac{4nL}{t+1} + (\frac{2}{t+2})^2 nL \tag{90}$$

$$= \frac{4nL}{t+2}\frac{t}{t+1} + (\frac{2}{t+2})^2 nL \leq \frac{4nL}{t+2}\frac{t+1}{t+2} + (\frac{2}{t+2})^2 nL \tag{91}$$

$$= \frac{4(t+2)nL}{(t+2)^2} = \frac{4nL}{t+2} = \frac{4nL}{(t+1)+1} \tag{92}$$

$\star$ Choose the line search step size $\mu_g^{(t)} = min\left(\frac{g(F^{(t)})}{L\|\partial\mathcal{H}^{*(t)} - F^{(t)}\|_F}, 1\right)$. Consider $\mathcal{Q}(F^{(t)})$ where

$$\mathcal{Q}(F^{(t)}) = \mathcal{M}(F^{(t)}) - \mu^{(t)}g(F^{(t)}) + (\mu^{(t)})^2 \frac{L}{2}\|\partial\mathcal{H}^{*(t)} - F^{(t)}\|_F^2 \tag{93}$$

Taking the derivative of $\mathcal{Q}(F^{(t)})$ with respect to $\mu^{(t)}$ and setting it equal to zero, we obtain:

$$\nabla_{\mu^{(t)}}\mathcal{Q}(F^{(t)}) = \frac{\partial}{\partial\mu^{(t)}}\left(\mathcal{M}(F^{(t)}) - \mu^{(t)}g(F^{(t)}) + (\mu^{(t)})^2\frac{L}{2}\|\partial\mathcal{H}^{*(t)} - F^{(t)}\|_F^2\right) \tag{94}$$

$$= -g(F^{(t)}) + \mu^{(t)}L\|\partial\mathcal{H}^{*(t)} - F^{(t)}\|_F^2 = 0 \tag{95}$$

We can obtain that:

$$\mu^{(t)} = \frac{g(F^{(t)})}{L\|\partial\mathcal{H}^{*(t)} - F^{(t)}\|_F^2} \tag{96}$$

Since $\mu^{(t)}$ is defined as the convex combination coefficient between $F^{(t)}$ and $\partial\mathcal{H}^{*(t)}$, we have $\mu^{(t)} \leq 1$, specifically:

$$\mu^{(t)} = \min\left(\frac{g(F^{(t)})}{L\|\partial\mathcal{H}^{*(t)} - F^{(t)}\|_F}, 1\right) \tag{97}$$

This is the definition of $\mu_g^{(t)}$. This means that choosing $\mu_g^{(t)}$ always minimizes $\mathcal{Q}$ as much as possible. Given that $F^{(t+1)}$ is updated using the step size $\mu_g^{(t)}$, and $\tilde{F}^{(t+1)}$ is updated using the simple step

size, we have:

$$\mathcal{M}(F^{(t+1)}) \leq \mathcal{M}(F^{(t)}) - \mu_g^{(t)} g(F^{(t)}) + (\mu_g^{(t)})^2 \frac{L}{2} \|\partial \mathcal{H}^{*(t)} - F^{(t)}\|_F^2 \tag{98}$$

$$\leq \mathcal{M}(F^{(t)}) - \mu_e^{(t)} g(F^{(t)}) + (\mu_e^{(t)})^2 \frac{L}{2} \|\partial \mathcal{H}^{*(t)} - F^{(t)}\|_F^2 \tag{99}$$

$$\leq (1 - \mu_e^{(t)}) \mathcal{M}(F^{(t)}) + (\mu_e^{(t)})^2 nL \leq \frac{4nL}{t+1} \tag{100}$$

Thus, we have fully proved that for any of the three step sizes $\{\mu_e, \mu_l, \mu_g\}$, the algorithm will converge to the global optimum $F^*$, with a convergence rate of $\frac{4nL}{t+1}$.

### A.7  PROOF FOR THEOREM 4.10.

Assume that $min_{F \in \Omega} \mathcal{H} \in P_{DB}^L$ and that $\mathcal{H}$ has a global minimum $F^*$. $\tilde{g}^{(t)}$ represents the smallest dual gap $g^{(t)}$ obtained during the first $t$ iterations of the DNF algorithm, i.e., $\tilde{g}^{(t)} = \min_{1 \leq k \leq t} g^{(k)}$. By using $\mu_g^{(t)}$ as step. Then $\tilde{g}^{(t)}$ satisfies the following inequality:

$$\tilde{g}^{(t)} \leq \frac{\max\{2(\mathcal{H}(F^{(0)}) - \mathcal{H}(F^*)), 2nL\}}{\sqrt{t+1}} \tag{101}$$

*Proof.* In this theory, we assume the problem to be solved is $\min_{F \in \Omega} \mathcal{H} \in P_{DB}^L$, meaning that $\mathcal{H}$ only needs to satisfy $L$-smoothness without requiring full differentiability. This implies the following conditions:

$$\mathcal{H}(F^{(t+1)}) \leq \mathcal{H}(F^{(t)}) + \langle \nabla \mathcal{H}(F^{(t)}), (F^{(t+1)} - F^{(t)}) \rangle + \frac{L}{2} \|F^{(t+1)} - F^{(t)}\|_F^2 \tag{102}$$

By definition, let $F^{(\mu)} = F^{(t)} + \mu^{(t)} d^{(t)}$, where $d^{(t)} = \partial \mathcal{H}^{(t)} - F^{(t)}$.

$$\mathcal{H}(F^{(\mu)}) \leq \mathcal{H}(F^{(t)}) + \mu \langle \nabla \mathcal{H}(F^{(t)}), d^{(t)} \rangle + \frac{L}{2} (\mu)^2 \|F^{(\mu)} - F^{(t)}\|_F^2 \tag{103}$$

$$\leq \mathcal{H}(F^{(t)}) + \mu \langle \nabla \mathcal{H}(F^{(t)}), d^{(t)} \rangle + \frac{L}{2} \sup_{\forall F^{(\mu)}, F^{(t)} \in \Omega} \|F^{(\mu)} - F^{(t)}\|_F^2 \tag{104}$$

$$\leq \mathcal{H}(F^{(t)}) - \mu g^{(t)} + (\mu)^2 Ln. \tag{105}$$

At this point, assume the upper bound function $\mathcal{B} = -\mu g^{(t)} + (\mu)^2 Ln$. Consider taking the partial derivative of $\mathcal{B}$ with respect to $\mu$:

$$\frac{\partial \mathcal{B}}{\partial \mu} = -g^{(t)} + 2\mu Ln = 0 \Rightarrow \mu = \frac{g^{(t)}}{2Ln} \tag{106}$$

To choose the step size that minimizes $\mathcal{B}$, we set $\mu^{(t)} = \min\left\{\frac{g^{(t)}}{2Ln}, 1\right\}$. Let $\mathcal{I}_{[\cdot]}$ denote an indicator function. Specifically, $\mathcal{I}_{[g^{(t)} > 2Ln]}$ is defined as:

$$\mathcal{I}_{[g^{(t)} > 2Ln]} = \begin{cases} 1, & \text{if } g^{(t)} > 2Ln, \\ 0, & \text{otherwise.} \end{cases} \tag{107}$$

Thus, we have:

$$\mathcal{H}(F^{(t+1)}) \leq \left(\mathcal{H}(F^{(t)}) - \mu g^{(t)} + (\mu)^2 Ln\right)\big|_{\mu=\mu^{(t)}} \mathcal{H}(F^{(t)}) - \mathcal{B}\left(\min\left\{\frac{g^{(t)}}{2Ln}, 1\right\}\right) \tag{108}$$

$$= \mathcal{H}(F^{(t)}) - \left(\frac{(g^{(t)})^2}{4nL} \mathcal{I}_{[g^{(t)} \leq 2Ln]} + (g^{(t)} - nL) \mathcal{I}_{[g^{(t)} > 2Ln]}\right) \tag{109}$$

$$= \mathcal{H}(F^{(t)}) - \min\left(\frac{(g^{(t)})^2}{4nL}, (g^{(t)} - nL) \mathcal{I}_{[g^{(t)} > 2Ln]}\right) \tag{110}$$

Summing both sides of the inequality from $t = 0$ to $t = T$, we obtain:

$$\sum_{t=0}^{T} \mathcal{H}(F^{(t+1)}) \leq \sum_{t=0}^{T} \mathcal{H}(F^{(t)}) - \sum_{t=0}^{T} \min\left( \frac{(g^{(t)})^2}{4nL}, (g^{(t)} - nL)\mathcal{I}_{[g^{(t)}>2Ln]} \right) \tag{111}$$

Rearranging terms and simplifying:

$$\mathcal{H}(F^{(T+1)}) - \mathcal{H}(F^{(0)}) \leq - \sum_{t=0}^{T} \min\left( \frac{(g^{(t)})^2}{4nL}, (g^{(t)} - nL)\mathcal{I}_{[g^{(t)}>2Ln]} \right) \tag{112}$$

It is easy to verify that the following equation obviously holds:

$$\sum_{t=0}^{T} \min\left( \frac{(g^{(t)})^2}{4nL}, (g^{(t)} - nL)\mathcal{I}_{[g^{(t)}>2Ln]} \right) \leq (T+1) \min\left( \frac{(g^{(t)})^2}{4nL}, (g^{(t)} - nL)\mathcal{I}_{[g^{(t)}>2Ln]} \right) \tag{113}$$

we have the following:

$$\mathcal{H}(F^{(T+1)}) - \mathcal{H}(F^{(0)}) \leq -(T+1) \min\left( \frac{(g^{(t)})^2}{4nL}, (g^{(t)} - nL)\mathcal{I}_{[g^{(t)}>2Ln]} \right) \tag{114}$$

$$\leq -(T+1) \min\left( \frac{\tilde{g}^2}{4nL}, (\tilde{g} - nL)\mathcal{I}_{[\tilde{g}>2Ln]} \right) \tag{115}$$

Where $\tilde{g}^{(t)}$ represents $\min_{1 \leq k \leq T} g^{(k)}$, which is the smallest dual gap within $T$ steps. At this point, we need to discuss which case $\tilde{g}$ falls into within $\min\left( \frac{\tilde{g}^2}{4nL}, (\tilde{g} - nL)\mathcal{I}_{[\tilde{g}>2Ln]} \right)$.

$\star$ If $\tilde{g} \leq 2nL$, we have $\mathcal{H}(F^{(T+1)}) - \mathcal{H}(F^{(0)}) \leq -(T+1)\frac{\tilde{g}^2}{4nL}$, By simplifying, we can obtain an upper bound for $\tilde{g}$ as:

$$\tilde{g} \leq \sqrt{\frac{4nL(\mathcal{H}(F^{(0)}) - \mathcal{H}(F^{(T+1)}))}{T+1}} \leq \sqrt{\frac{4nL(\mathcal{H}(F^{(0)}) - \mathcal{H}(F^*))}{T+1}} \tag{116}$$

Where $F^*$ is the global optimal point of $\mathcal{H}$, and $\mathcal{H}(F^*)$ is the global minimum of $\mathcal{H}(F)$.

$\star$ If $\tilde{g} \geq 2nL$, we have $\mathcal{H}(F^{(T+1)}) - \mathcal{H}(F^{(0)}) \leq -(T+1)(\tilde{g} - nL)$. By simplifying, we can obtain an upper bound for $\tilde{g}$ as $\tilde{g} \leq nL + \frac{\mathcal{H}(F^{(0)}) - \mathcal{H}(F^*)}{T+1}$, at that time, we have:

$$2Ln \leq \tilde{g} \leq nL + \frac{\mathcal{H}(F^{(0)}) - \mathcal{H}(F^*)}{T+1} \Rightarrow T+1 \leq \frac{\mathcal{H}(F^{(0)}) - \mathcal{H}(F^*)}{nL} \tag{117}$$

In summary, we obtain:

$$\tilde{g} \leq \begin{cases} \sqrt{\frac{4nL(\mathcal{H}(F^{(0)}) - \mathcal{H}(F^*))}{T+1}}, & \text{if} \tilde{g} \leq 2nL, \\ nL + \frac{\mathcal{H}(F^{(0)}) - \mathcal{H}(F^*)}{T+1}, & \text{otherwise.} \end{cases} \tag{118}$$

and we have (Lacoste-Julien, 2016):

$$nL + \frac{\mathcal{H}(F^{(0)}) - \mathcal{H}(F^*)}{T+1} = \frac{\mathcal{H}(F^{(0)}) - \mathcal{H}(F^*)}{\sqrt{T+1}} \left( \frac{nL}{\mathcal{H}(F^{(0)}) - \mathcal{H}(F^*)}\sqrt{T+1} + \frac{1}{\sqrt{T+1}} \right) \tag{119}$$

$$\leq \frac{\mathcal{H}(F^{(0)}) - \mathcal{H}(F^*)}{\sqrt{T+1}} \left( \frac{1}{\sqrt{T+1}} + \sqrt{\frac{nL}{\mathcal{H}(F^{(0)}) - \mathcal{H}(F^*)}} \right) \tag{120}$$

$$\leq \frac{\mathcal{H}(F^{(0)}) - \mathcal{H}(F^*)}{\sqrt{T+1}} \left( \frac{1}{\sqrt{T+1}} + 1 \right) \tag{121}$$

$$\leq \frac{2(\mathcal{H}(F^{(0)}) - \mathcal{H}(F^*))}{\sqrt{T+1}} \tag{122}$$

The first inequality holds because $\frac{\mathcal{H}(F^{(0)}) - \mathcal{H}(F^*)}{nL} > T + 1$, and the second inequality holds because $\mathcal{H}(F^{(0)}) - \mathcal{H}(F^*) \leq nL$. So we have:

$$\tilde{g} \leq \begin{cases} \sqrt{\frac{4nL(\mathcal{H}(F^{(0)}) - \mathcal{H}(F^*))}{T+1}}, & \text{if } \tilde{g} \leq 2nL, \\ 2\frac{\mathcal{H}(F^{(0)}) - \mathcal{H}(F^*)}{\sqrt{T+1}}, & \text{otherwise.} \end{cases} \tag{123}$$

Since $\sqrt{4nL\big(\mathcal{H}(F^{(0)}) - \mathcal{H}(F^*)\big)} \leq \max\{2\big(\mathcal{H}(F^{(0)}) - \mathcal{H}(F^*)\big), 2nL\}$, it follows that:

$$\tilde{g} \leq \frac{\max\{2\big(\mathcal{H}(F^{(0)}) - \mathcal{H}(F^*)\big), 2nL\}}{\sqrt{T+1}} \tag{124}$$

Proof completed.

A.8  PROOF FOR THEOREM 4.11.

For $F^{(t)} \in \Omega$ and convex function $\mathcal{H}$, $g(F^{(t)}) \geq \mathcal{H}(F^{(t)}) - \min_{F \in \Omega} \mathcal{H}(F) = \mathcal{H}(F^{(t)}) - \mathcal{H}(F^*)$, and when $g^{(t)}$ converges to 0 at $\mathcal{O}(\frac{1}{T})$, it means that $\mathcal{H}(F^{(t)}) - \min_{F \in \Omega} \mathcal{H}(F) = \mathcal{H}(F^{(t)}) - \mathcal{H}(F^*) \to 0$ at $\mathcal{O}(\frac{1}{T})$. More generally, if $\mathcal{H}$ is not a convex function, then $g(F^{(t)}) = 0$ if and only if $F^{(t)}$ is a stable critical point of $\mathcal{H}$.

*Proof.* For $F^{(t)} \in \Omega$ and convex function $\mathcal{H}$, we have:

$$g(F^{(t)}) \geq \mathcal{H}(F^{(t)}) - \min_{F \in \Omega} \mathcal{H}(F) = \mathcal{H}(F^{(t)}) - \mathcal{H}(F^*) \tag{125}$$

where $F^*$ is the global minimizer of $\mathcal{H}$. When $g^{(t)}$ converges to 0 at the rate $\mathcal{O}(\frac{1}{T})$, it implies:

$$\mathcal{H}(F^{(t)}) - \min_{F \in \Omega} \mathcal{H}(F) = \mathcal{H}(F^{(t)}) - \mathcal{H}(F^*) \to 0 \tag{126}$$

at the rate $\mathcal{O}(\frac{1}{T})$. This proof is identical to the previous one.

For the second part, which states: If $\mathcal{H}$ is not a convex function, then $g(F^{(t)}) = 0$ if and only if $F^{(t)}$ is a stable critical point of $\mathcal{H}$, we have the following: If $g(F^{(t)}) = 0$, this means that the gradient $\nabla \mathcal{H}(F^{(t)})$ has a non-positive inner product with the feasible domain $\Omega$, implying that the direction within the feasible domain is always an ascent direction. Thus, $F^*$ must be a stable point. The reverse proof is similar.

A.9  PROOF FOR THEOREM 4.13.

For size constrained min cut, its line search step size $\mu_l^{(t)}$ has an analytical solution $\mu_l^{*(t)}$.

*Proof.* Since this analysis holds for each iteration of running the DNF algorithm, we abbreviate $\mu_l^{(t)}$ as $\mu_l$, and the update rule is written as $F \leftarrow (1 - \mu_l)F + \mu_l \partial \mathcal{H}$, where $\mu_l$ is obtained by $\mu_l = \arg\min_{\mu \in (0,1)} \mathcal{H}\big((1 - \mu)F + \mu \partial \mathcal{H}\big)$ Substituting into the loss function of Min-Cut, we have:

$$\mu_l^* = \arg\min_{\mu \in (0,1)} -\text{tr}\Big(\big((1 - \mu)F + \mu \partial \mathcal{H}\big)^T S \big((1 - \mu)F + \mu \partial \mathcal{H}\big)\Big) \tag{127}$$

Now, expanding the expression inside the trace:

$$\text{tr}\Big(\big((1 - \mu)F + \mu \partial \mathcal{H}\big)^T S \big((1 - \mu)F + \mu \partial \mathcal{H}\big)\Big) = \text{tr}\big((1 - \mu)^2 F^T S F + 2\mu(1 - \mu)F^T S \partial \mathcal{H} + \mu^2 (\partial \mathcal{H})^T S \partial \mathcal{H}\big) \tag{128}$$

Thus, the formula becomes:

$$\mu_l^* = \arg\min_{\mu \in (0,1)} \big(-\text{tr}\big((1 - \mu)^2 F^T S F\big) - 2\mu(1 - \mu)\text{tr}\big(F^T S \partial \mathcal{H}\big) - \mu^2 \text{tr}\big((\partial \mathcal{H})^T S \partial \mathcal{H}\big)\big) \tag{129}$$

For the expression

$$\left(-\mathrm{tr}\left((1-\mu)^2 F^T S F\right) - 2\mu(1-\mu)\mathrm{tr}\left(F^T S \partial \mathcal{H}\right) - \mu^2 \mathrm{tr}\left((\partial \mathcal{H})^T S \partial \mathcal{H}\right)\right) \tag{130}$$

simplifying this expression leads to the standard quadratic form:

$$\mu_l^* = \arg\max_{\mu \in (0,1)} \quad \alpha^2(x + y - 2z) + 2\alpha(z - y) + y \tag{131}$$

where $\alpha = 1 - \mu$, $x = \mathrm{tr}\left(F^T S F\right)$, $y = \mathrm{tr}\left((\partial \mathcal{H})^T S \partial \mathcal{H}\right)$, $z = \mathrm{tr}\left((\partial \mathcal{H})^T S F\right)$.

Consider $\alpha^2(x + y - 2z) + 2\alpha(z - y)$

⋆ If $x + y - 2z \leq 0$, the parabola opens downward, and we have:

$$\mu_l^* = \begin{cases} 1 - \frac{y-z}{x+y-2z}, & \text{if}\frac{y-z}{x+y-2z} \in (0,1), \\ 0, & \text{if}\frac{y-z}{x+y-2z} \geq 1, \\ 1, & \text{if}\frac{y-z}{x+y-2z} \leq 0. \end{cases} \tag{132}$$

⋆ If $x + y - 2z \geq 0$, the parabola opens upward, and we have:

$$\mu_l^* = \begin{cases} 1, & \text{if}|1 - \frac{y-z}{x+y-2z}| \geq |\frac{y-z}{x+y-2z}|, \\ 0, & \text{if}|1 - \frac{y-z}{x+y-2z}| \leq |\frac{y-z}{x+y-2z}|. \end{cases} \tag{133}$$

In conclusion, we can directly obtain the line search result for the DNF method for the min-cut problem without actually performing the search.

## B  RELATED WORKS AND TECHNICAL DETAILS

### B.1  GRAPH CLUSTERING

To resolve the imbalance clustering results in MC, several normalization criteria have been introduced, such ratio cut (Rcut) (Chan et al., 1993), normalized cut (Ncut) (Wan et al., 2024) and min-max cut (Ding et al., 2001). Each method employs a unique normalization approach to balance the partition sizes and improve clustering quality. In Ruct, the normalization involves dividing by the size of the sub-clusters, while in Ncut, the normalization factor is the sum of degrees of the nodes within the respective clusters. The min-max cut further enhances the approach by simultaneously minimizing inter-cluster similarity and maximizing intra-cluster compactness. By incorporating normalization terms, these methods reformulate the optimization problem into the spectral clustering framework, which include eigenvalue decomposition on the graph Laplacian and subsequent K-Means (KM) discretization. KM also suffers from imbalanced clustering results due to the optimization. Some balanced regularization terms could be added in KM or MC to aviod skewed results, ensuring clusters are well-distributed and meaningful (Chen et al., 2019). For instance, the fast clustering with flexible balance constraints (FCFC) and balanced KM with novel constraint (BKNC) are proposed for balanced clustering results (Liu et al., 2018; Chen et al., 2022). The fast adaptively balanced MC clustering method is presented by adding balanced factors (Nie et al., 2025). To more intuitively avoid trivial solutions, (Nie et al., 2024) propose size constrained MC, which adds size constrains on each cluster to avoid small-sized clusters. However, the optimization problem is difficult to solve effectively. In this paper, we relaxed the indicator matrix and resolved the problem from the perspective of non-linear optimal transport.

### B.2  OPTIMAL TRANSPORT

Optimal transport (OT) theory has recently received significant interest because of its versatility and wide-ranging applications across numerous fields. (Villani, 2003) established the mathematical foundation of OT, offering a powerful framework for measuring distances between target and source distributions. (Kantorovich, 2006) relaxed the original problem of Monge. The convex linear program optimization determines an optimal matching, minimizing the cost of transferring mass between two distributions. (Cuturi, 2013) revolutionize the field by introducing the Sinkhorn algorithm (Sinkhorn & Knopp, 1967), which employs entropy regularization to make the computation of optimal transport more efficient and scalable to high-dimensional data. (Peyré et al., 2019b) further develop algorithms for computational optimal transport, enhancing its practicality for large-scale problems. The Gromov-Wasserstein (GW) distances (Mémoli, 2011) generalizes OT to scenarios where the ground spaces are not pre-aligned, resulting in a non-convex quadratic optimization problem for transport computation. (Peyré et al., 2016) extends GW distances and derive a fast entropically-regularized iterative algorithm to access the stationary point. However, the bound is generally fixed. (Shi et al., 2024a) relaxed the bound into flexible ones and propose doubly bonded OT problem and applied it into partition-based clustering. Nontheless, they merely solves the linear convex problem. In this paper, we concentrate on double-bounded nonlinear OT problem and apply it in size constrained MC clustering.

### B.3  IMPLEMENTATION NOTES

During the iteration process, the selected stopping condition for the iteration is 500 iterations. In practical applications, one can calculate the value of the gap function and then choose the point closest to 0 as the final optimized point. In the visualization of $F$, the values of the elements in $F$ are recombined, with samples from the same cluster arranged together. The visualizations of indicator matrices in this paper follow similar operations. The implementation of DNF is publicly accessible at `https://anonymous.4open.science/r/DNF_code-3FD0`.

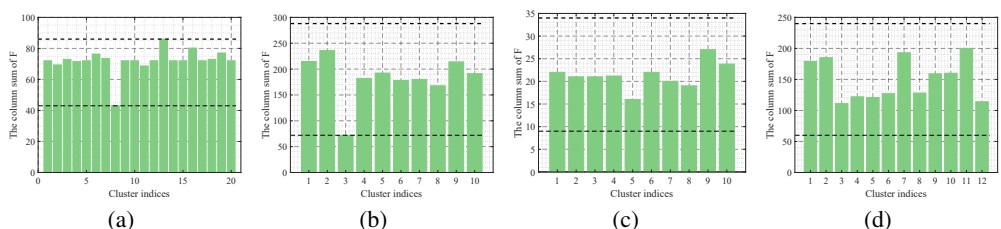

Figure 5: The clustering distribution with lower and upper bounds. (a) COIL20. (b) Digit. (c) JAFFE. (d) MSRA25.

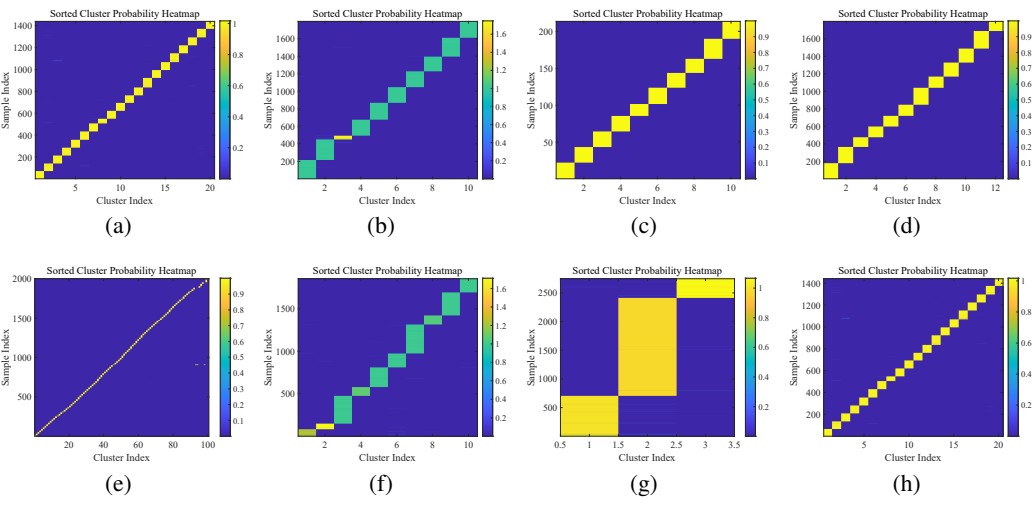

Figure 6: The visualization of obtained indicator matrix of size constrained MC problem solved by DNF on real datasets. (a) COIL20. (b) Digit. (c) JAFFE. (d) MSRA25. (e) PalmData25. (f) USPS20. (g) Waveform21. (h) MnistData05.

## C  ADDITIONAL RESULTS

### C.1  ADDITIONAL CLUSTERING DISTRIBUTION

The final clustering distributions and indicator matrices for each dataset are visualized in Figures 5 and 6, respectively. By comparing Figure 2 and 5, it can be seen that after applying DNF to the size-constrained method, the number of samples in each cluster of the final clustering result satisfies the constraint, which also indicates the validity of the solution obtained by DNF. Since the indicator matrix is arranged in order, Figure 6 shows that the clustering result exhibits a clear diagonal structure.

### C.2  TOY EXAMPLE

We visualized the comparison results of the proposed algorithm and the KM method on four toy datasets, including the Flame dataset, the Two ring dataset, and two custom-made datasets. The results are shown in Figure 7. It can be observed that, compared to KM, the proposed method is able to capture the local structure information of the data and achieves completely correct results on multiple datasets, demonstrating better clustering performance.

### C.3  DNF FOR CONVEX PROBLEM

In the theoretical analysis, we proved that DNF converges to the global optimal solution at a rate of $1/t$ when solving convex problems. Therefore, in this section, we use DNF to solve the following

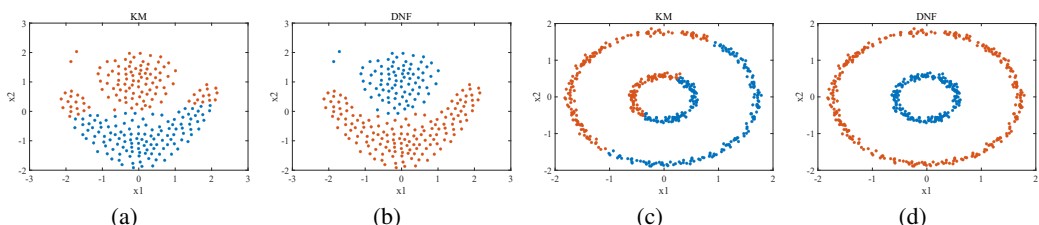

(a)       (b)       (c)       (d)

Figure 7: Visualization of KM and DNF applied in toy datasets. (a)-(b) Flame dataset. (c)-(d) Two ring dataset.

size constrained min cut problem.

$$\min_{F} Tr(F^T L F)$$
$$s.t. F1_c = 1_n, F \geq 0 \tag{134}$$

The solution to problem 134 is that all elements in the indicator matrix are equal to $1/c$. Figure 8 visualizes the changes in the indicator matrix and the objective function value with respect to the number of iterations on real datasets. The results in the figure show that the final indicator matrix does not clearly indicate the clustering structure of the samples. This also suggests that when the objective function is a convex problem in clustering, the solution will result in equal probabilities for a sample belonging to each cluster, which is invalid. In other words, clustering models with convex objective functions are problematic. Additionally, as seen from the number of iterations, when the optimization problem is convex, the objective function converges within 20 steps, which is consistent with the theoretical analysis and demonstrates that the DNF method has good convergence when solving convex problems.

### C.4 GAP FUNCTION VALUES

In the convergence analysis of Section 6.2, we plotted the changes in the objective function over iterations, and also recorded the changes in the objective function value of the gap function over iterations. The results of these changes are shown in Figure 9. It can be observed that during the iteration process, the value of the gap function continuously changes. The value of the gap function approaching 0 indicates that this point is the closest to the critical point. Therefore, in practice, the value of the gap function can be used to locate the optimal point.

### C.5 SENSITIVITY ABOUT $k$

Here, we analyze the effect of the number of neighbors $k$ on the clustering metrics ACC, NMI, and ARI. The results are shown in Figure 10. $k$ is a key parameter in constructing the nearest neighbor graph, with its value ranging from {6, 8, ..., 16}. As seen in Figure 10, the clustering results fluctuate on some datasets as $k$ changes. However, the fluctuation does not exceed 20%. Additionally, the fluctuation range is very small on datasets like COIL20 and PalmData25. In practice, it is recommended to set $k$ to 10 as an empirical value when using the algorithm.

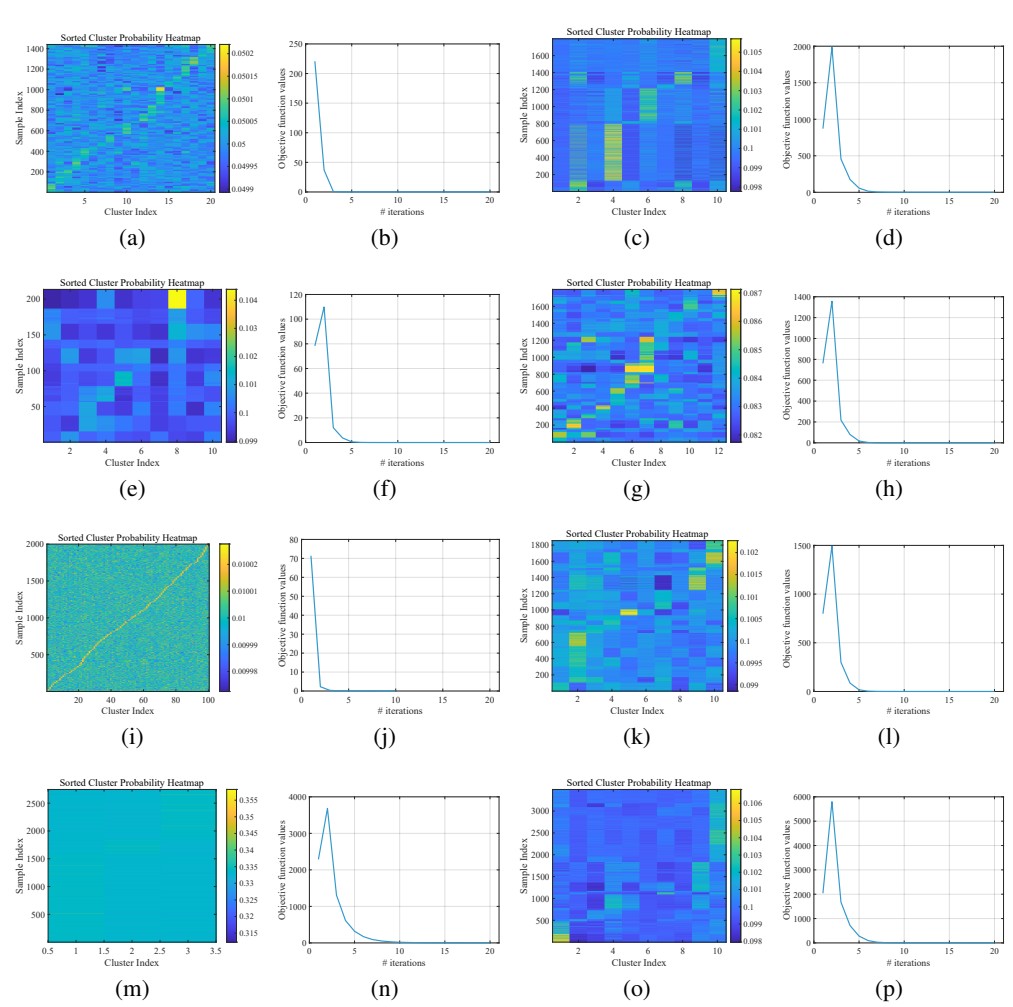

Figure 8: The visualization of obtained indicator matrix and variation of objective function values of min cut problems solved by DNF on real datasets. (a)-(b) COIL20. (c)-(d) Digit. (e)-(f) JAFFE. (g)-(h) MSRA25. (i)-(j) PalmData25. (k)-(l) USPS20. (m)-(n) Waveform21. (o)-(p) MnistData05.

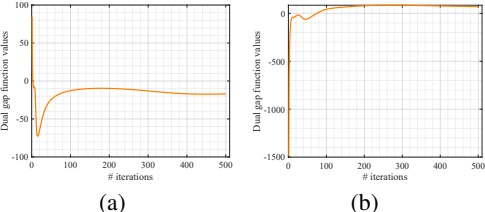

Figure 9: Variation of gap function values with the number of iterations. (a) PalmData25. (b) MnistData05.

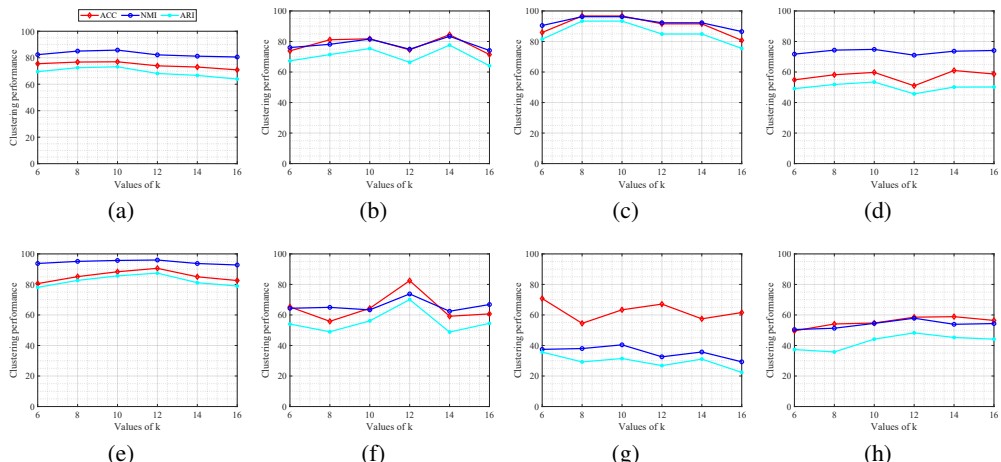

Figure 10: The sensitivity of the number of nearest neighbors $k$. (a) COIL20. (b) Digit. (c) JAFFE. (d) MSRA25. (e) PalmData25. (f) USPS20. (g) Waveform21. (h) MnistData05.

