# OpenReview forum: "Double-Bounded Nonlinear Optimal Transport for Size Constrained Min Cut Clustering"
_ICLR.cc/2026/Conference — ICLR 2026 Conference Withdrawn Submission_

### Official Review · Reviewer_Ut2a · 2025-10-24

**Soundness:** 1
**Presentation:** 1
**Contribution:** 1
**Rating:** 0
**Confidence:** 5

**Summary:**

This paper presents a Frank-world method for a doubly-bounded optimal transport problem, and attempts to apply it to the NP-hard problem of size-constrained minimum cut in graphs.

The paper claims to prove convergence guarantees for their algorithm under different conditions, and also claims that it applies to finding optimal solutions for the size-constrained minimum cut problem.

The proposed algorithm is compared in practice to several other clustering techniques on eight real world datasets.

**Strengths:**

The topic of a the paper (size constrained minimum cut in graphs) is an interesting and well motivated problem. The high-level idea of relaxing and NP-hard problem and then solving the relaxation to would, in principle, be a reasonable heuristic for getting approximate solutions.

**Weaknesses:**

Starting with the most crucial fundamental issue of the paper:

This paper is claiming to provide a polynomial time algorithm for the NP-hard problem of size-constrained graph cuts. The paper has itself acknowledged that this problem is NP-hard (line 136). If this were true then the title of the paper should be "A proof that P = NP". However, this is not the case. The algorithm appears to be a heuristic for graph clustering.

I wondered at first if perhaps the writing was just very unclear, making it seem like the paper was claiming to provide a poly-time solution for an NP-hard problem when that was not actually what it was trying to claim. This is especially because the paper goes back and forth between just talking about the "min-cut" problem (which, when this term is used correctly, refers to a poly-time graph cut problem) and the size-constrained min cut problem (which is NP-hard). However, it appears that throughout the manuscript that the authors are almost always using these terms interchangeably. And there are many places where it seems clear the paper is actually claiming to provide a poly-time solution for the NP-hard size-constrained minimum cut problem.

 Consider the following lines:

* Line 288: "For size constrained min cut, it is also modeled as a double-bounded nonlinear optimal transport problem, and is applicable to the DNF method" --> This line comes right after a section that claims the DNF method finds optimal solutions to optimal transport problems. Hence, it amounts to a claim that the (poly-time) DNF algorithm optimally solves and NP-hard problem. It also establishes that this subsection (and Theorem 3.12, which comes right after it), is referring to the size-constrained min cut problem and not the standard unconstrained min cut problem.

* Line 293: Theorem 3.12, says "By solving the minimum cut problem using the DNF algorithm, after t steps, the best step within t steps always converges to the optimal solution..." --> Although this theorem says "min cut", the context makes it clear the theorem is referring to the size-constrained min-cut problem, and hence this theorem is claiming to be optimally solving it using an iterative method that the paper later derives a polynomial runtime for.

* Line 306, "Furthermore, *we provide the algorithmic process for* solving general bilateral nonlinear optimal transport problems using the DNF method, as well as *the process for solving the size constrained min cut problem*" --> again, a claim of optimally solving an NP-hard problem.

* Line 085: "DNF method can achieve globally optimality..." --> This line comes after previous lines about how the method is used to solve the size-constrained min cut problem. In the same paragraph, it mentions a convergence rate of $O(1/\sqrt{t})$ in some settings, matching the convergence rate it claims in the abstract holds for the size-constrained min-cut problem.

* Line 092: "The size constrained min cut clustering framework benefits from the ability of DNF method to handle non-linear constraints effectively, *ensuring clusters of appropriate sizes while minimizing the cut value*"

Unfortunately these claims constitute a fundamental flaw in the paper. This and other aspects of the paper point to some apparent fundamental misunderstandings about the min-cut and size-constrained min cut problems:

* As noted before, the paper is not very clear in distinguishing between min cut and size constrained min cut. Most of the time, these appear to be used interchangeably
* There are also sometimes where the paper appears to realize a difference between min-cut and size constrained min-cut, but then conflate and confuse aspects of the two problems. The abstract opens by stating that "current solutions to the min cut problem suffer from slow speeds, difficulty in solving, and often converge to simple solutions." It is true that solving a simple min-cut problem often yields a trivial cut, but algorithms for this problem are not slow or difficult to solve.
* In the introduction, the paper claims that the MC problem has a trivial solution of not cutting any edges (here it appears to be talking about the unconstrained MC problem), and then uses this to support a claim that "Thus,...the clustering result of MC tends to produce unbalanced clusters..." But these are two different things. Yes, if you only minimize the cut value while allowing a trivial 'cluster' of size zero, you just wouldn't make a cut. But the standard min cut problem requires both clusters to be non-empty to avoid this trivial situation, so issues with imbalanced clusters are not somehow because of a tendency to have a cut value of zero.

Because of this, the paper cannot be published even with a significant re-write. I'd encourage the authors to carefully consider what they are proving and what they are trying to claim in this paper. In a best case scenario, perhaps the method proposed is solving some type of relaxation to an NP-hard problem, somehow converging to some sort of locally optimal solution, or maybe amounts to a heuristic that is not solving anything optimally but has decent results in practice.

**Questions:**

Can you confirm that it was your intent to claim that DNF is finding optimal solutions for an NP-hard problem?

---

### Official Review · Reviewer_3pen · 2025-10-29

**Soundness:** 2
**Presentation:** 2
**Contribution:** 2
**Rating:** 2
**Confidence:** 4

**Summary:**

To address the slow convergence to trivial solutions, the easy convergence to trivial solutions, and the unbalanced clustering problems of minimum-cut clustering, this paper transforms the scale-constrained minimum-cut problem into a double-bounded nonlinear optimal transport problem. It proposes a DNF algorithm based on the Frank-Wolfe method. The paper theoretically proves the convergence rate of DNF for convex functions satisfying Lipschitz smoothness, and for non-convex Lipschitz smooth functions. Experiments compare DNF with baseline methods on eight real-world datasets (including images, handwriting, and waveform data). DNF achieves the best results in both clustering accuracy and convergence speed.

**Strengths:**

1. The double-bounded nonlinear Frank-Wolfe (DNF) algorithm, proposed for the DB-NOT problem, offers significant advantages in optimization logic and detailed design. Compared to traditional methods, DNF requires no manual parameter tuning and can be directly adapted to different datasets. Experiments have demonstrated superior stability. It also offers two gradient approximation strategies: the "norm basis" and the "inner product basis," which can be selected based on the data size.
2. The paper transforms the minimum cut clustering problem with scale constraints into a double-boundary nonlinear optimal transmission problem, while expanding the scope of application of classical optimal transmission.

**Weaknesses:**

1. The time complexity of DNF is \(O(n(m+1)c)\) (n is the number of samples, m is the number of nonzero elements in each row of the similarity matrix S, and c is the number of clusters). When n reaches the "million level", matrix multiplication will experience excessive memory usage and a sharp drop in iteration speed. However, the paper neither tested this scenario nor proposed solutions such as parallel computing and distributed optimization.
2. The paper addresses the cluster imbalance problem in MC using "double boundaries" but does not provide a specific selection strategy for \(b_l\) and \(b_u\). In practical applications, users need to set them based on experience, and there is no analysis of the impact of \(b_l \) and \ (b_u\) on clustering results.
3. While the comparative methods include Scut and other methods, they do not include double-boundary OT-related algorithms or compare deep learning-based MC methods, resulting in a lack of comprehensive horizontal comparison for the conclusion that "DNF has the best performance".

**Questions:**

1. How to choose the double boundary (\(b_l, b_u\))? Is it universal across different datasets?

---

### Official Review · Reviewer_FoRu · 2025-11-01

**Soundness:** 2
**Presentation:** 2
**Contribution:** 2
**Rating:** 2
**Confidence:** 4

**Summary:**

The paper considers the min-cut clustering problem by relaxing the discrete indicator matrix into a probabilistic form with double-bounded size constraints. It then proposes a Double-Bounded Nonlinear Optimal Transport formulation and introduces the Double-Bounded Nonlinear Frank-Wolfe (DNF) algorithm to efficiently solve it.

**Strengths:**

The problem setting is interesting, the solver is practical, and the theoretical guarantees are appropriate for a Frank‑Wolfe style method.

**Weaknesses:**

See questions below.

**Questions:**

1. In the experimental section, the main objective seems to be demonstrating that the proposed min-cut formulation performs competitively with other clustering methods. However, this formulation appears to have already been introduced by Nie et al. (2024). Could you clarify what is new compared with their work? If the key novelty lies in the proposed Frank-Wolfe-based optimization scheme, then comparing it against alternative optimization algorithms would be a more appropriate evaluation.
2. Page 2. "We prove that the DNF method can achieve global optimality regardless of whether the nonlinear function is convex or Lipschitz-continuous non-convex." However, in the paper it is only showed converges to a critical point.
3. Could you specify in the experiment setting, how to you choose b_l and b_u?
4. For proof of the convergence rate in the Theorem 3.9/Theorem A.8, it claims $H(F^{(t)})-H(F^*)<4L/(t+1)$, but in the proof what is actually showed is $4nL/(t+1)$?
5. Line 175, why the overall feasible set $\Omega=\Omega_1 \cup \Omega_2 \ cup \Omega_3$ it should be the intersection instead?

---

### Official Review · Reviewer_ioSv · 2025-11-02

**Soundness:** 2
**Presentation:** 2
**Contribution:** 2
**Rating:** 2
**Confidence:** 3

**Summary:**

The paper introduces DB-NOT, a double-bounded nonlinear optimal transport formulation, and DNF, a Frank–Wolfe-style solver. The method is instantiated for size-constrained min-cut clustering and evaluated on eight datasets. The theory gives rates for convex and nonconvex cases; experiments show improvements over several classical baselines. Claims about optimality for nonconvex objectives and the relation to prior double-bounded OT work require clarification and stronger evidence.

**Strengths:**

1. Convergence analysis with rates O(1/t) (convex, L-smooth) and O($1/\sqrt{t}$) (nonconvex, L-smooth).
2. A general constrained OT viewpoint that could transfer to tasks beyond clustering.
3. Public code and an evaluation on eight benchmark datasets.

**Weaknesses:**

1. Prior work on double-bounded OT [1] is closely related; the incremental novelty of the nonlinear extension is not established and there is no direct comparison in formulation, algorithmic choices, or results.
2. The abstract and main text conflate convergence to optimality in the convex case with nonconvex guarantees. Theorem statements provide convergence to a critical point for nonconvex settings, not global optimality; the language in the abstract should be corrected.
3. Empirical validation focuses on mid-scale data and classical baselines; large-scale tests and modern baselines are missing. No evidence of scalability to large n (e.g., $n \geq 10^5$); no stochastic or mini-batch variant; no wall-clock or memory breakdown versus scalable OT or graph clustering methods.
4. Sensitivity to key construction choices (e.g., k-NN graph, Gaussian bandwidth) and any entropic regularization or step-size scheme is not ablated.
5. Baselines omit modern deep and GNN-based clustering methods, limiting the empirical scope.
6. No ablation for graph construction parameters or step-size choices.
7. No guidance for choosing bl, bu or handling cases where feasible bounds conflict with graph structure.

[1] Shi, Liangliang, Zhaoqi Shen, and Junchi Yan. “Double-bounded optimal transport for advanced clustering and classification.” Proceedings of the AAAI Conference on Artificial Intelligence, 38(13), 2024.

**Questions:**

1. How does DNF scale to $n > 10^5$? Is there a stochastic or mini-batch variant, and what are the per-iteration time and memory costs with sparse graphs?
2. Why is there no direct comparison with double-bounded OT (AAAI 2024) in objectives, constraints, and performance?
3. How are the lower and upper size bounds $b_l, b_u$ selected across datasets? Is there an automatic rule or a validation strategy?
4. For nonconvex objectives, what precise statement is supported by Theorem 3.10 regarding stationarity vs. optimality, and will the abstract be revised accordingly?

---

### Note · Authors · 2025-11-12

I have read and agree with the venue's withdrawal policy on behalf of myself and my co-authors.